# Frictional Q-Learning

**Hyunwoo Kim** [1]   **Hyo Kyung Lee** [1]

## Abstract

Off-policy reinforcement learning suffers from extrapolation errors when a learned policy selects actions that are weakly supported in the replay buffer. In this study, we address this issue by drawing an analogy to static friction. From this perspective, the replay buffer is represented as a smooth, low-dimensional action manifold, where the support directions correspond to the tangential component, while the normal component captures the dominant first-order extrapolation error. This decomposition reveals an intrinsic anisotropy in value sensitivity that naturally induces a stability condition analogous to a friction threshold. To mitigate deviations toward unsupported actions, we propose Frictional Q-Learning, an off-policy algorithm that encodes supported actions as tangent directions using a contrastive variational autoencoder. We further show that an orthonormal basis of the orthogonal complement corresponds to normal components under mild local isometry assumptions. Extensive empirical results on standard continuous-control benchmarks consistently demonstrate robust and stable performance compared with competitive baselines.

## 1. Introduction

Off-policy reinforcement learning trains agents using a replay buffer that aggregates past interactions with the environment (Watkins et al., 1989). However, when the policy queries state–action pairs that are absent from or underrepresented in the replay buffer, extrapolation error arises, leading to inaccurate policy updates (Thrun & Schwartz, 2014). To address this problem, Batch Constrained Q-learning (BCQ) (Fujimoto et al., 2019) restricts learned policies to remain close to the visitation distribution of the replay buffer via a variational autoencoder (VAE) (Kingma & Welling, 2013).

Specifically, when the policy remains within the support of the replay buffer, which grounds value estimates in observed data, extrapolation error is effectively mitigated. Locally, the support concentrates on a smooth low-dimensional manifold, which is well approximated by its tangent space (Roweis & Saul, 2000). Accordingly, tangent components characterize perturbations that remain consistent with the support. In contrast, normal components correspond to off-support perturbations that induce maximal first-order departure (Petersen, 2006). Consequently, these normal components pull the policy away from support, and the resulting actions fail to satisfy the constraints.

To formalize this limitation, we draw an analogy between extrapolation error and static friction. In classical mechanics, static friction opposes the tangential component of a force (such as gravity) and prevents downward motion along the slope toward the equilibrium state at the horizontal surface. As the inclination angle increases, the required static friction also increases and eventually reaches its maximum threshold. Analogously, extrapolation error acts as a resistance to the convergence of the policy toward the ground truth. As the distributional mismatch between the replay buffer and the environment widens, this error intensifies, hindering effective learning (Gulcehre et al., 2020). Just as the slope angle governs the balance between tangential and normal force components, the ratio between the tangential and normal gradients of the extrapolation error quantifies the relative resistance to policy updates. Therefore, to overcome this resistance, the policy must prioritize actions aligned with the tangent directions of the manifold while strictly avoiding normal components. This enforces a support-based constraint that effectively suppresses extrapolation error.

We introduce *Frictional Q-learning* (FQL), an off-policy algorithm that employs a contrastive variational autoencoder (cVAE) (Abid & Zou, 2019) to generate candidate actions under dual objectives: (i) alignment with the tangent directions and (ii) separation from the normal component in the action manifold. This contrastive structure explicitly identifies the supported action and generates candidates that satisfy the batch constraint more reliably than standard algorithms. To implement these dual constraints, we assume a stability condition that bounds the sensitivity of the critic to perturbations and demonstrate that an orthonormal basis of the orthogonal complement provides an effective projection

[1]Data Analytics and Healthcare Systems Lab, Korea University, Seoul, Korea. Correspondence to: Hyo Kyung Lee <hyokyunglee@korea.ac.kr>.

of normal directions. Given the bounded nature of the action space, we apply an affine transformation to recenter the coordinates, ensuring candidates are evaluated within the feasible domain. This process yields a finite set of candidates representing normal directions, which serve as background samples for the contrastive structure via the critic network. We evaluate the proposed method on standard continuous-control benchmarks from MuJoCo environments (Todorov et al., 2012; Towers et al., 2024) and demonstrate that it achieves competitive performance relative to off-policy and offline baselines. This work is the first to introduce a geometric interpretation of batch RL grounded in a physical analogy. We expect our perspective to open avenues for developing policy classes and to serve as a foundation for future research. Our code is available on GitHub ⌗.

## 2. Related Work

Since the performance of deep off-policy RL is closely tied to the alignment between the replay buffer and the test distribution (Mnih et al., 2015; Isele & Cosgun, 2018), BCQ (Fujimoto et al., 2019) introduced a *batch-constrained* approach to mitigate distributional shifts. Formally, a policy is strictly batch-constrained if and only if every selected state–action pair resides within the support of the replay buffer, thereby eliminating extrapolation error. A state-conditioned generative network approximates the behavioral distribution by restricting candidate actions to those with high likelihood given observed samples. Under standard Robbins–Monro conditions (Robbins & Monro, 1951), the batch-constrained policy converges to the optimal value function of the empirical deterministic Markov Decision Process (MDP), and its fixed point coincides with that of the optimal policy constrained to the batch. Ideally, such a policy ensures coherence, meaning that all induced trajectories remain fully contained within the support of the replay buffer. Consequently, the batch-constrained policy offers greater stability than any unconstrained alternatives when initialized from a state within the replay buffer. Related efforts include BEAR (Kumar et al., 2019), which employs Maximum Mean Discrepancy (MMD) to constrain the learned policy within the support of the behavior distribution, and BAIL (Chen et al., 2020), which selects a subset of high-value transitions to facilitate imitation learning while avoiding out-of-distribution regions. While these regularized approaches effectively prevent unrealistic value estimates for unseen state–action pairs, they face a critical theoretical limitation: the bound on extrapolation error scales with $(1-\gamma)^{-2}$. This quadratic scaling implies that as the discount factor $\gamma$ approaches 1, the error bound becomes excessively loose (Xi et al., 2021). Thus, in long-horizon MDPs, even a slight distributional discrepancy can yield significantly inflated theoretical errors, potentially leading to an overestimation of the expected return (Wu et al., 2022; Liu et al., 2023).

## 3. Background

We consider an MDP $\mathcal{M} = (\mathcal{S}, \mathcal{A}, p_{\mathcal{M}}, r, \gamma)$ with state space $\mathcal{S}$, action space $\mathcal{A}$, transition dynamics $p_{\mathcal{M}}$, reward function $r$, and discount factor $\gamma \in [0, 1)$. At timestep $t$, the agent observes $s$, selects $a$, receives reward $r = r(s, a, s') \in \mathbb{R}$, and transitions to $s' \sim p_{\mathcal{M}}(\cdot|s, a)$. The objective is to learn a policy $\pi(a|s)$ that maximizes the discounted return $R_t = \sum_{i=t}^{\infty} \gamma^{i-t} r(s_i, a_i, s_{i+1})$. A policy $\pi$ induces a visitation distribution $\mu^{\pi}$ over $\mathcal{S}$. The Bellman operator $\mathcal{T}^{\pi}$ is a $\gamma$-contraction and admits a unique fixed point, action-value function $Q^{\pi}(s, a) = \mathbb{E}[R_t|s, a]$. The Bellman optimality operator converges to the optimal action-value function $Q^{\star}(s, a) = \max_{\pi} Q^{\pi}(s, a)$. The greedy action $a^{\star} = \arg\max_a Q^{\star}(s, a)$ recovers an optimal policy (Bertsekas, 2008). In large state spaces, the true action-value function $Q^{\pi}$ is typically approximated by a neural network. However, greedy actions become intractable in continuous action spaces; actor–critic methods are commonly used (Konda & Tsitsiklis, 1999). The critic $Q_{\varphi}(s, a)$ is trained using a temporal-difference target

$$y = r + \gamma Q_{\varphi^-}\big(s', \pi_{\omega^-}(s')\big),$$

while the actor $\pi_{\omega}(s)$ is updated with respect to this value estimate via the policy gradient (Silver et al., 2014):

$$\nabla_{\omega} J(\omega) = \mathbb{E}_{s \sim \mu^{\pi_{\omega}}} \big[ \nabla_a Q_{\varphi}(s, \pi_{\omega}(s)) \big| \nabla_{\omega} \pi_{\omega}(s) \big].$$

To regulate overestimation, both target networks $Q_{\varphi^-}$ and $\pi_{\omega^-}$ are updated via a $\xi$-weighted ($0 < \xi \ll 1$) moving average of the current parameters and their previous target values (Lillicrap et al., 2015).

### 3.1. Extrapolation error

In batch RL, the agent observes only a finite set of transitions $(s, a, r, s')$ in the replay buffer $\mathcal{B}$ (Lin, 1992). Based on this dataset, we construct an empirical MDP $\mathcal{M}_{\mathcal{B}}$ that shares the identical state, action, and reward spaces as the true MDP $\mathcal{M}$. The transition dynamics of $\mathcal{M}_{\mathcal{B}}$ are given by $p_{\mathcal{B}}(s'|s, a) = \frac{N(s, a, s')}{\sum_{\tilde{s}} N(s, a, \tilde{s})}$, where $N(s, a, s')$ denotes the count of transitions $(s, a, s')$ in $\mathcal{B}$. For any unobserved state–action pair, the transition deterministically terminates in a synthetic state $s_{\text{init}}$ (i.e., $p_{\mathcal{B}}(s_{\text{init}}|s, a) = 1$), and its reward $r(s, a, s_{\text{init}})$ is assigned as the initial value estimate $Q_{\mathcal{M}}(s, a)$ to avoid undefined Bellman targets. We define the tabular *extrapolation error* $\mathcal{E}_{\mathcal{B}}(s, a)$ as the discrepancy between the value function $Q_{\mathcal{B}}$, computed under the empirical MDP $\mathcal{M}_{\mathcal{B}}$ induced by the batch $\mathcal{B}$, and the value function $Q$ of the true MDP $\mathcal{M}$. For any policy $\pi$, this error admits the following Bellman-like recursive formulation:

$$\mathcal{E}_\mathcal{B}(s,a) = Q^\pi_\mathcal{M}(s,a) - Q^\pi_\mathcal{B}(s,a)$$
$$= \sum_{s'} \big(p_\mathcal{M}(s'|s,a) - p_\mathcal{B}(s'|s,a)\big)$$
$$\Big(r(s,a,s') + \gamma \sum_{a'} \pi(a'|s')Q^\pi_\mathcal{B}(s',a')\Big) \quad (1)$$
$$+ \sum_{s'} p_\mathcal{M}(s'|s,a)\gamma \sum_{a'} \pi(a'|s')\mathcal{E}_\mathcal{B}(s',a').$$

Formally, extrapolation error arises from a mismatch between the dataset distribution $\mu_\mathcal{B}(s,a)$ and the state–action distribution induced by the learned policy. In standard off-policy Q-learning (Precup et al., 2001), Bellman backups rely solely on transitions $(s,a,r,s') \in \mathcal{B}$. However, the policy may query subsequent state–action pairs $(s', \pi(s'))$ that lie outside the support of the replay buffer, or are only weakly represented. In such cases, Bellman updates are forced to rely on unsupported value estimates, requiring the evaluation of actions favored by the learned policy but not sufficiently represented in $\mathcal{B}$. This leads to biased and increasingly inaccurate $Q$-values via bootstrapping (Kumar et al., 2019). Furthermore, since absent state–action pairs cannot be reconstructed via density-based reweighting, adjusting the loss under the current policy cannot address errors when the policy assigns probability mass to actions outside the dataset.

### 3.2. Static Friction

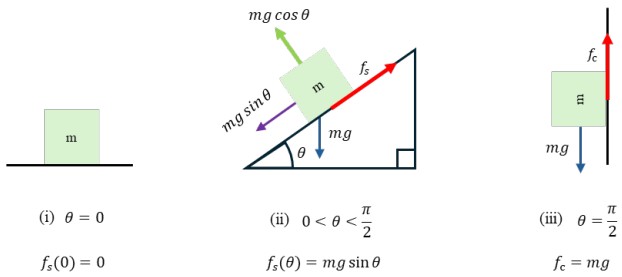

(i) $\theta = 0$
$f_s(0) = 0$

(ii) $0 < \theta < \dfrac{\pi}{2}$
$f_s(\theta) = mg\sin\theta$

(iii) $\theta = \dfrac{\pi}{2}$
$f_c = mg$

*Figure 1.* Consider a body of mass $m$ resting on a plane inclined at an angle $\theta$ relative to the horizontal. The gravitational force $mg$ acts vertically downward and can be decomposed into two components with respect to the plane: a tangential component and a normal component perpendicular to the surface (Newton, 1833).

Static friction is a contact force that resists relative tangential motion between solid surfaces. It arises when the surfaces remain at rest and increases proportionally to the applied force until it reaches a threshold, beyond which motion begins. In Figure 1, three cases are considered: (i) at $\theta = 0$, no tangential component exists, and thus no static friction is exerted. (ii) For $0 < \theta < \pi/2$, the tangential component $mg\sin\theta$ acts downslope, and static friction $f_s$ opposes it.

Under the Coulomb model, the static friction is bounded by

$$f_s \le f_{s,\max} = \mu_s N = \mu_s\, mg\cos\theta.$$

At the threshold, the static friction reaches its maximum value and perfectly balances the tangential component:

$$f_{s,\max} = mg\sin\theta = \mu_s N, \quad \mu_s = \tan\theta. \quad (2)$$

Beyond this threshold, any further increase in $\theta$ initiates motion, at which moment kinetic friction takes over. In case (iii) where $\theta = \pi/2$, the normal force vanishes ($N = 0$), and static friction cannot be sustained. In this configuration, an external compressive force $f_c = mg$ must support the entire weight to prevent downward motion. The coefficient of static friction $\mu_s$ is a constant determined solely by the surfaces and quantifies the ability of the interface to resist tangential force. Overall, static friction provides a bounded resistance that opposes the natural tendency of the body to move toward the most gravitationally stable configuration.

### 3.3. Analogy

We formalize the analogy in which extrapolation error $\mathcal{E}_\mathcal{B}(s,a)$ exhibits a threshold structure analogous to the Coulomb relation in Equation (2). From this analogy, we derive an anisotropy ratio based on the tangent and normal directions of the action manifold $M_\mathcal{B}$. The validity of this decomposition rests on a standard batch-constraint assumption: at each state $s$, the actions in the dataset concentrate on a locally low-dimensional manifold, and the encoder-decoder pair $(E_{M_\mathcal{B}}, D_{M_\mathcal{B}})$ of the generative network provides a smooth reconstruction map that approximates this state-conditional support.

### 3.4. Action Manifold

In the regime where the reconstruction error of the decoder is negligible, the tangent space of the action manifold $T_s M_\mathcal{B}$ captures the first-order directions that preserve reconstruction fidelity (Petersen, 2006). For an infinitesimal perturbation $\varepsilon \in \mathbb{R}^d$, the stability condition for reconstruction is

$$D_{M_\mathcal{B}}\big(s, E_{M_\mathcal{B}}(s, a + \varepsilon)\big) = a + O(\|\varepsilon\|^2).$$

This condition holds provided that the perturbation lies within the tangent space. Conversely, perturbations along the normal space $N_s M_\mathcal{B}$ induce a reconstruction distortion of first-order $O(\|\varepsilon\|)$, corresponding to unsupported, off-manifold deviations. Thus, the tangent space induced by the generative network provides a first-order local approximation of the supported directions around $a$. Consequently, the generative network defines a manifold where tangent directions are supported, whereas its normal directions characterize unsupported deviations. In Figure 2, unit vectors $t \in T_s M_\mathcal{B}$ and $n \in N_s M_\mathcal{B}$ (where $t \perp n$) define the plane

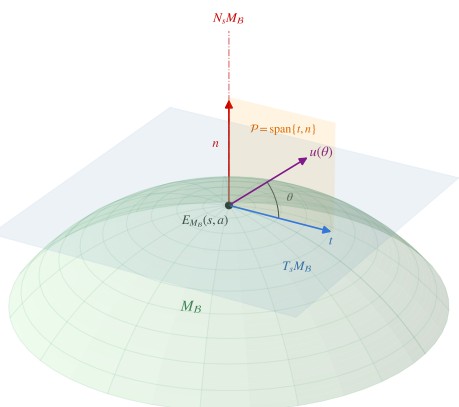

*Figure 2.* In a smooth action manifold $M_\mathcal{B}$, unit vectors of subspace $T_s M_\mathcal{B}$ and $N_s M_\mathcal{B}$ span the space $\mathcal{P}$ with unit direction $u(\theta)$. Note that $u(\theta)$ lies entirely within the 2D plane $\mathcal{P}$ spanned by the tangent and normal vectors. The angle $\theta$ explicitly denotes the rotation of $u$ within this coplanar space.

$\mathcal{P} = \text{span}\{t, n\}$. The unit direction $u(\theta) = \cos\theta\, t + \sin\theta\, n$ is parameterized by the angle $0 \leq \theta < \pi/2$, which quantifies the relative alignment with respect to the support of the replay buffer.

### 3.5. Frictional Constraint

Under standard smoothness conditions, the Bellman operator is locally Lipschitz-continuous with respect to the action $a$ (Munos & Szepesvári, 2008). Hence, the extrapolation error $\mathcal{E}_\mathcal{B}(s, a)$ exhibits local directional variation which is linearly bounded with respect to directional Lipschitz factors $L_u$. For a small perturbation $\varepsilon u(\theta)$ induced by the perturbation of $a$, the variation in extrapolation error is characterized by a directional finite difference. The directional growth of the extrapolation error $g_{u(\theta)}(s, a)$ can be characterized in terms of the tangential and normal directions

$$g_{u(\theta)}(s, a) := \limsup_{\varepsilon \to 0} \frac{\left| \mathcal{E}_\mathcal{B}(s, a + \varepsilon u(\theta)) - \mathcal{E}_\mathcal{B}(s, a) \right|}{\varepsilon}.$$

For $u(\theta) = \cos\theta\, t + \sin\theta\, n$, the directional growth satisfies

$$g_{u(\theta)}(s, a) \leq g_t(s, a) \cos\theta + g_n(s, a) \sin\theta.$$

For a batch-constrained policy, we focus on updates that remain close to the support that attains the smallest directional growth in the tangential direction, where the associated Lipschitz bound is minimal among all unit directions:

$$g_t(s, a) = \min_{\|u\|=1} g_u(s, a), \quad L_t = \min_{\|u\|=1} L_u. \quad (3)$$

The anisotropy ratio $\kappa$ characterizes the relative directional growth of extrapolation error around the action manifold. We therefore define the tolerance $\lambda_{\text{tol}}$ as the ratio between the normal and tangential directional Lipschitz bounds. This tolerance $\lambda_{\text{tol}}$ bounds the $|\kappa| \tan(\theta)$ to define the region where value evaluation remains reliable near the support:

$$\kappa = \frac{g_n(s, a)}{g_t(s, a)}, \quad \lambda_{\text{tol}} = \frac{L_n}{L_t}, \quad |\kappa| \tan\theta \leq \lambda_{\text{tol}}.$$

The tolerance $\lambda_{\text{tol}}$ ensures local regularity of the value function and prevents error amplification from off-support directions. This condition implies the existence of a critical ratio beyond which deviations are no longer controlled near the support (Asadi et al., 2018). Importantly, the tolerance $\lambda_{\text{tol}}$ is not a hyperparameter and does not impose explicit directional Lipschitz constraints. While global Lipschitz bounds can be approximately enforced under restrictive architectural assumptions, directional Lipschitz control in deep neural networks is generally infeasible in practice (Gouk et al., 2021; Anil et al., 2019). Instead, it serves as an analytical quantity that upper bounds the combined effect of the anisotropy ratio $\kappa$ and the local angle $\theta$:

$$\theta \leq \arctan\left( \frac{\lambda_{\text{tol}}}{|\kappa|} \right), \quad \frac{1}{|\kappa|} = \left| \frac{g_t(s, a)}{g_n(s, a)} \right|. \quad (4)$$

Equation (4) requires the tangential error growth to remain sufficiently small relative to the normal error growth in the local manifold. A *small* value of the reciprocal indicates that perturbations along the tangent direction induce substantially less extrapolation error than those along the normal direction, and the policy naturally favors the tangent space.

## 4. Algorithm

Since there are infinitely many vectors in the normal space, it is difficult to select optimal samples that maximize the normal gradient of the extrapolation error. Therefore, we show that the orthonormal basis of the orthogonal complement can be associated with a normal direction of the action manifold. An orthonormal basis provides a deterministic set of normal directions that span the normal space. These normal directions yield the strongest contrast and render $v$ a feasible and informative background sample under an affine transformation for the action space. The replay buffer is assumed to be fully supported by the tangent directions.

### 4.1. Action Geometry

For a batch-constrained optimal policy, the latent representation $u := E_{M_\mathcal{B}}(s, a)$ retains dominant first-order tangential properties and provides a first-order approximation of the local tangent space $T_s M_\mathcal{B}$. A unit vector $w$ then lies in the normal space $N_s M_\mathcal{B}$ if it satisfies $w^\top u = 0$. To relate orthogonality in the action manifold to a sufficient orthogonality condition in the action space, we first consider an idealized linear isometric encoder $E_{M_\mathcal{B}}$, which preserves the local geometric structure of the action space (Gropp et al., 2020; Lee et al., 2022). To understand latent-space

orthogonality, consider first a linear encoder $E(x) = Wx$ with an orthogonal matrix $W$ ($W^\top W = I$). For an action $a$ and a perturbation vector $v$ such that $v \perp a$, the inner product in the latent space is strictly preserved:

$$\langle Wa, Wv \rangle = a^\top W^\top W v = a^\top I v = a^\top v = 0.$$

This intuition extends to nonlinear encoders through local linearization. Given that the encoder is differentiable, let $J(s, a)$ denote the Jacobian of the encoder. A first-order Taylor expansion around $(s, a)$ yields

$$E(s, a + \varepsilon v) \approx E(s, a) + \varepsilon J(s, a) v.$$

Here, the latent perturbation vector $c$ corresponds to the directional derivative, defined as $c := J(s, a)v$. Under the local isometry assumption, where $J(s, a)^\top J(s, a) \approx I$ in high-density regions (Gropp et al., 2020; Lee et al., 2022), the inner product between the latent representations is

$$\begin{aligned} c^\top u &\approx (J(s, a)v)^\top (J(s, a)a) \\ &= v^\top \left( J(s, a)^\top J(s, a) \right) a \\ &\approx v^\top a = 0. \end{aligned}$$

Thus, any unit direction orthogonal to the action constructs a normal direction on the manifold through the isometric encoder, ensuring that perturbations in these directions induce maximal deviation from the learned tangent space.

**Theorem 4.1.** *Let $v \in \mathbb{R}^d$ be a unit vector satisfying $v^\top a = 0$. Under the local isometry assumption on the encoder $E_{M_B}$, the latent representation $c = E_{M_B}(s, v)$ satisfies $c^\top u = 0$ to first order, where $u = E_{M_B}(s, a)$.*

However, when the action manifold satisfies $2 \leq \dim(M_B) \ll d$, the latent representation $c$ is not strictly orthogonal to every tangent direction. Nevertheless, $u$ remains the dominant direction within the local tangent space. As a result, the residual tangent component of $c$ along the action manifold is negligible (Appendix A.1), and $c$ remains associated with the normal space even in higher dimensions.

### 4.2. Affine Transformation

The orthogonal complement of the action $a$, denoted by $a^\perp \subset \mathbb{R}^d$, admits an orthonormal basis set $V = \{v_1, \ldots, v_{d-1}\}$. By Theorem 4.1, the latent representation of each vector $c_i = E_{M_B}(s, v_i)$ lies in the normal space $N_s M_B$ to first order, and each $c_i$ is interpreted as a normal direction relative to the local manifold. However, it is difficult to construct orthogonal vectors $v_i$ that simultaneously lie inside the bounded action space $\mathcal{A} = [x, y]^d$. We leverage the fact that intermediate action directions need not remain strictly within the bounded domain. To address this, we apply an affine transformation to recenter the action space and compute an orthonormal basis of the orthogonal

complement within this transformed space $\mathcal{A}'$. Then, the selected actions are mapped back to the original action space via the inverse affine transformation for interaction with the environment.

Let $\mathcal{A} = [x, y]^d$ be a bounded action space and apply the following affine transformation to obtain the space $\mathcal{A}'$:

$$\tilde{a}_i = \frac{a_i - \mathcal{S}}{r}, \qquad \mathcal{S} = \frac{x + y}{2}, \quad r = \frac{y - x}{2}.$$

Since $v_i$ is a unit vector, we have $v_i \in \mathcal{A}'$ for all $i$. This affine transformation yields a deterministic set of unit vectors, thereby avoiding the ambiguity inherent in the infinite number of normal-component candidates. Therefore, they can be paired with distinct state–action tuples for augmentation. In addition, as the directions in $V$ remain mutually orthogonal, so probes $(s, v_i)$ and $(s, v_j)$ capture independent normal-space perturbations and provide informative samples for contrast.

### 4.3. Stability Bound

For an MDP $\mathcal{M}_B$ and a reward function $|r(s, a, s')| \leq R_{\max}$, the extrapolation error $\mathcal{E}_\theta(s, a)$ in Equation (1) satisfies the following upper bound (Appendix A.2):

$$\left| \mathcal{E}_\theta \right|_\infty \leq C \sup_{s, a} \Delta(s, a),$$

$$C = \frac{2R_{\max}}{(1 - \gamma)^2}, \quad \Delta(s, a) = \mathrm{TV}\big(p_0(s'|s, a), p_\theta(s'|s, a)\big).$$

Under local Lipschitz regularity, the directional growth of the total variation $\delta(s, a)$ satisfies

$$\delta_{u(\theta)}(s, a) := \limsup_{\varepsilon \to 0} \frac{\left| \Delta(s, a + \varepsilon u(\theta)) - \Delta(s, a) \right|}{\varepsilon}.$$

Since the extrapolation error is Lipschitz with respect to $\Delta(s, a)$, its directional growth is bounded by

$$g_{u(\theta)}(s, a) \leq C \, \delta_{u(\theta)}(s, a).$$

On the other hand, Equation (3) implies that the anisotropy ratio $\kappa$ satisfies the bound for a batch-constrained policy:

$$\frac{1}{|\kappa|} = \left| \frac{g_t(s, a)}{g_n(s, a)} \right| \leq 1.$$

Thus, the stability condition in Equation (4) is expressed as

$$\tan \theta \leq \frac{\lambda_{\mathrm{tol}}}{|\kappa|} \leq \lambda_{\mathrm{tol}}. \tag{5}$$

Equation (5) shows that the local angle $\theta$ is governed by the relative directional sensitivity between tangential and normal components, whereas the factor $(1 - \gamma)^{-2}$ does not affect the stability bound. The range of policy deviations is not fixed a priori by the Lipschitz constant; instead, the anisotropy ratio $\kappa$ imposes a stricter constraint on $\theta$. Crucially, the directional stability constraint on $\theta$ is strictly tighter than a bound based solely on a global tolerance.

## 4.4. Contrastive Variational Autoencoder

We define the target dataset as $\mathcal{D}_t := \{x_i = (s_i, a_i)\}_{i=1}^n$ and construct a background dataset $\mathcal{D}_b := \{b_i = (s_i, v_i^*)\}_{i=1}^n$ where $v_i$ is an element of the orthonormal basis set $V$ obtained from the affine transformation. Since the replay buffer stores the set $V$ and the state $s_i$, we select the background sample $v_i^*$ that yields the **lowest** estimated Q-value at every update. The contrastive variational autoencoder $G_\zeta$ consists of two Gaussian encoders, $q_\phi^{\bar{s}}$ and $q_\phi^z$, and a decoder $f_\eta$ (Abid & Zou, 2019). It decomposes the latent representation into *salient* factors $\bar{s} \sim \mathcal{N}(0, I)$, which capture variations unique to the target dataset, and *irrelevant* factors $z \sim \mathcal{N}(0, I)$, which model the structure shared with the background distribution. This is achieved by two coupled variational evidence lower bounds for a target sample $x_i$ and a background sample $b_i$:

$$\mathcal{L}_t(x_i) \leq \mathbb{E}_{q_\phi^{\bar{s}} q_\phi^z} \left[ \log f_\eta(a_i | s_i, \bar{s}, z) \right] \\ - \beta \left( \mathrm{KL}[q_\phi^{\bar{s}}(\bar{s}|x_i) \| p(\bar{s})] + \mathrm{KL}[q_\phi^z(z|x_i) \| p(z)] \right),$$

$$\mathcal{L}_b(b_i) \leq \mathbb{E}_{q_\phi^z} [\log f_\eta(v_i^* | s_i, 0, z)] - \beta\, \mathrm{KL}[q_\phi^z(z|b_i) \| p(z)].$$

Concretely, we regularize the joint posterior to factorize each feature as $(\bar{s}, z) = (q_\phi^{\bar{s}}(s, a), q_\phi^z(s, a))$ using the total correlation (TC) which equals 0 when $\bar{s}$ and $z$ are independent and increases otherwise (Chen et al., 2018):

$$\mathrm{TC} := \mathrm{KL}\left( q_\phi^{\bar{s}} q_\phi^z(\bar{s}, z | x_i) \,\|\, q_\phi^{\bar{s}}(\bar{s}|x_i) \cdot q_\phi^z(z|x_i) \right).$$

Since this KL term is not directly tractable under mini-batch estimation, we approximate it using the density-ratio trick (Kim & Mnih, 2018). A discriminator $D_\psi$ distinguishes between samples drawn from the joint distribution and those from the product of marginals, where the latter is approximated by randomly shuffled latent pairs within each batch:

$$\mathrm{TC} \approx \log \frac{D_\psi(\bar{s}, z)}{1 - D_\psi(\bar{s}, z)}.$$

Finally, the objective of the contrastive variational autoencoder with a hyperparameter $\beta \geq 0$ that balances KL divergence with reconstruction fidelity is:

$$\max_\zeta \frac{1}{n} \sum_{i=1}^n \left[ \mathcal{L}_t(x_i) + \mathcal{L}_b(b_i) - \mathrm{TC}(\bar{s}, z) \right].$$

This latent factorization implements the contrastive structure: supported actions induce the tangent space (salient), whereas normal directions populate the normal space (irrelevant) without a tangential component. Thus, when the decoder reconstructs action candidates using $(\bar{s}, 0)$, it becomes insensitive to nuisance structures, ensuring that only salient variations affect the generated actions. Consequently, $v_i$ acts as a principled, structured negative sample that separates irrelevant variation from the salient dynamics in the replay buffer. To ensure functional invariance, where an input of zero remains fixed throughout the computation, all biases are removed, and ReLU activations are employed.

---

**Algorithm 1** Frictional Q-learning (FQL)

Initialize parameter vectors $\varphi_1, \varphi_1^-, \varphi_2, \varphi_2^-, \omega, \omega^-, \zeta$
**for** each iteration **do**
  **for** each step **do**
    $\tilde{a} \sim f_\eta(s), \quad a = \pi_\omega(s, \tilde{a}) + \mathcal{N}$
    $s' \sim p_\mathcal{M}(s'|s, a), \quad v^* = \arg\min_{v \in V} Q_{\varphi_1}(s, v)$
    $\mathcal{B} \leftarrow \mathcal{B} \cup \{(s, a, r(s, a, s'), s', V)\}$
    $y = r + \gamma \min_{l=1,2} Q_{\varphi_l^-}(s', \pi_{\omega^-}(s', f_\eta(s')))$
    $\omega \leftarrow \arg\max_\omega \sum \left( Q_{\varphi_1}(s, \pi_\omega(s, \tilde{a})) \right)$
    $\varphi \leftarrow \arg\min_\varphi \sum \left( y - Q_\varphi(s, a) \right)^2$
    $\zeta \leftarrow \arg\max_\zeta \sum \left( \mathcal{L}_t(s, a) + \mathcal{L}_b(s, v^*) - \mathrm{TC}(\bar{s}, z) \right)$
    $\varphi_\ell^- \leftarrow \xi \varphi_\ell + (1 - \xi)\varphi_\ell^-, \omega^- \leftarrow \xi\omega + (1 - \xi)\omega^-$
  **end for**
**end for**

---

## 4.5. Frictional Q-learning

To preserve the constraint, we introduce a state-conditioned marginal density $P_\mathcal{B}(a|s)$ that represents the likelihood of an action $a$ at state $s$ under the replay buffer distribution. A policy aiming to maximize $\pi^\star(s) = \arg\max_a P_\mathcal{B}(a|s)$ restricts action selection to the supported space of the action manifold, thereby reducing extrapolation error from out-of-distribution state-action pairs. However, direct estimation of $P_\mathcal{B}(a|s)$ in high-dimensional continuous spaces is generally infeasible. As a practical alternative, we train a contrastive generative network $G_\zeta = \{q_\phi^{\bar{s}}, q_\phi^z, f_\eta\}$ to approximate this state-conditioned action distribution. Generated actions $\tilde{a} \sim f_\eta(s)$ from the decoder serve as surrogates for high-likelihood actions and are evaluated by the critic $Q_{\varphi_1}$:

$$\pi(s) = \arg\max_a Q_{\varphi_1}(s, a), \quad a = \{\pi_\omega(s, \tilde{a}_i)\}_{i=1}^m$$

A set of background candidates is evaluated using the critic network to identify the least-supported actions: the candidate with the lowest value estimate is selected as the background sample. In parallel, the random latent variables $\bar{s}, z$ are decoded with the current state to generate candidate actions. Then, the double Q-network structure (Hasselt, 2010; Fujimoto et al., 2018) selects the candidate with the highest value estimate. The overall process constitutes the Frictional Q-Learning algorithm; its specification is presented in Algorithm 1 and Appendix C. Our algorithm restricts value evaluation to a finite set of actions generated at each state. These actions either lie in the tangent space induced by the replay buffer or correspond to normal direction perturbations that are penalized by design. Consequently, the value function need not assign values to arbitrarily extrapolated actions. Instead, value updates remain confined to a controlled subset whose geometry is fixed by the data distribution. This restriction prevents unbounded value estimation and enables the policy to learn the dynamics toward a stable fixed point with formal convergence guarantees (Appendix A.3).

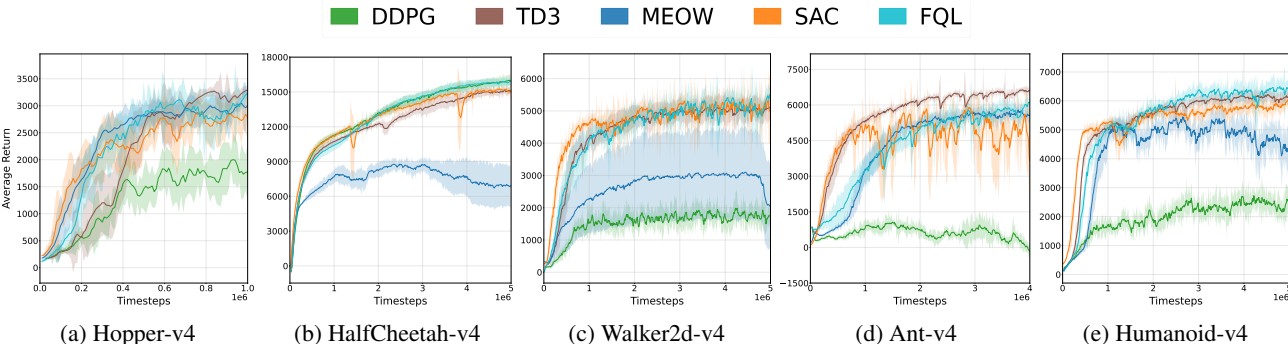

*Figure 3.* Average return (solid line) and half of the standard deviation (shaded area) across five independent experiments with different random seeds in continuous-control environments. For visual clarity, mean curves are smoothed with an exponential moving average.

*Table 1.* Average return of final evaluation over algorithms and environments. The best-performed value is marked in **bold**.

| Algorithm | Hopper-v4 | HalfCheetah-v4 | Walker2d-v4 | Ant-v4 | Humanoid-v4 |
|---|---|---|---|---|---|
| FQL | **3370.51 ± 220.37** | **15969.59 ± 554.20** | **5659.86 ± 510.08** | 6176.60 ± 207.67 | **6437.15 ± 722.05** |
| SAC | 2548.36 ± 850.06 | 15476.83 ± 244.64 | 4815.50 ± 1698.4 | 3098.49 ± 3690.32 | 5682.62 ± 327.95 |
| MEOW | 3190.29 ± 222.19 | 7241.62 ± 3722.44 | 1919.23 ± 2524.40 | 5635.11 ± 356.79 | 4650.21 ± 1168.05 |
| TD3 | **3371.87 ± 181.78** | 15120.68 ± 517.59 | 5104.41 ± 787.84 | **6542.34 ± 310.11** | 5373.53 ± 1879.89 |
| DDPG | 1824.07 ± 1061.24 | 15628.10 ± 981.48 | 1744.66 ± 436.61 | -237.80 ± 507.44 | 2618.13 ± 749.56 |

## 5. Results

Our algorithm is evaluated using continuous control tasks from MuJoCo (Todorov et al., 2012) within the Gymnasium benchmark suite (Towers et al., 2024), a widely adopted simulation platform for RL. We compare FQL against a diverse set of off-policy RL baselines: Deep Deterministic Policy Gradient (DDPG) (Lillicrap et al., 2015), Twin Delayed DDPG (TD3) (Fujimoto et al., 2018), Soft Actor–Critic (SAC) (Haarnoja et al., 2018), and Maximum Entropy Reinforcement Learning via Energy-Based Normalizing Flow (MEOW) (Chao et al., 2024). Additionally, we conduct offline RL experiments (Levine et al., 2020), in which algorithms are trained exclusively on fixed replay buffers, without further interaction with the environment, to compare our method with batch-constrained Q-learning (BCQ) (Fujimoto et al., 2019). *Imitation learning* (Vecerik et al., 2017; Hester et al., 2018) is also considered, where DDPG constructs a replay buffer from suboptimal or moderate behavior policies. Under these conditions, the target algorithm induces an action manifold that accentuates sensitivity to extrapolation error. Furthermore, the background dataset is augmented with a basis set $V$ to assess robustness to off-support perturbations. Each algorithm is trained independently using an NVIDIA RTX A6000. For each environment, we report the average episode return. Baselines were reproduced using Cleanrl (Huang et al., 2022) and their official implementations. Hyperparameter configurations of experimental results are detailed in Appendix B.

## 5.1. Comparison

We compared primarily against standard baselines without such auxiliary architectural modifications to isolate the intrinsic geometric advantages of our framework. As shown in Figure 3 and Table 1, the proposed method outperforms baseline approaches across multiple continuous-control tasks, including Hopper-v4, HalfCheetah-v4, Walker2d-v4, and Humanoid-v4. In environments such as Ant-v4, its performance is comparable to that of baseline methods. The strong performance observed on high-dimensional tasks indicates that the proposed method remains robust across various levels of action-space complexity and data coverage. Due to the stochasticity introduced by latent variables in the decoder, FQL exhibits exploration behavior comparable to that of stochastic policies while retaining the efficiency and stability of a deterministic framework. This stability is attributed to the explicit regulation of value estimates in unsupported regions of the action space, which suppresses harmful extrapolation. These frictional constraints are enforced in both the tangential and normal directions within the generative network. This performance advantage is not limited to locomotion benchmarks. In Appendix D.1, we further observe that the proposed method consistently outperforms or is comparable to the baseline on various manipulation benchmarks, including DMControl (Tassa et al., 2018) and MyoSuite (Caggiano et al., 2022), demonstrating that the benefits of our framework generalize beyond locomotion to broader continuous control tasks.

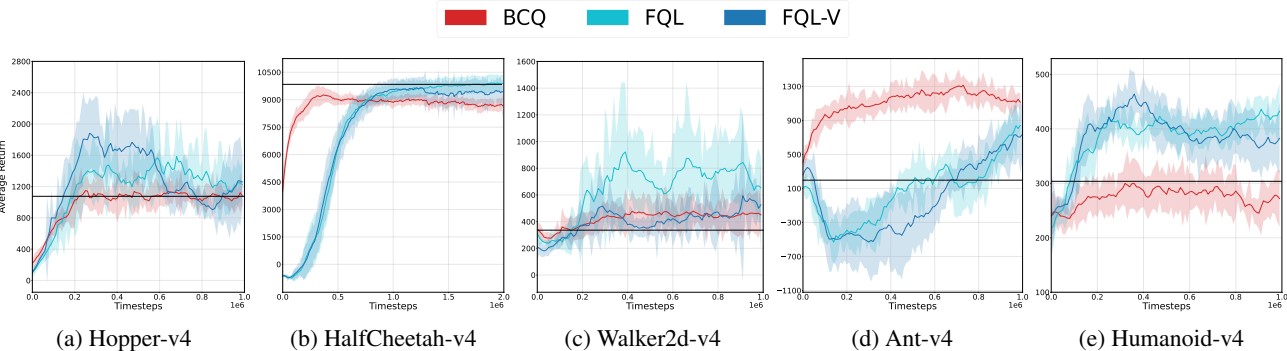

*Figure 4.* Average return (solid line) and half of the standard deviation (shaded area) of *imitation learning* across five independent experiments with different random seeds in continuous-control environments. The black line indicates the average episodic return for trajectories in the replay buffer. For visual clarity, mean curves are smoothed with an exponential moving average.

## 5.2. Imitation Learning

To further verify the frictional constraints, we evaluate the algorithm through *imitation learning*. In this regime, the policy is trained exclusively to reproduce actions contained in a fixed replay buffer, without additional exploration or online data collection (Levine et al., 2020). This design disentangles the intrinsic ability to regulate the support of the replay buffer from any performance gains arising from improved off-policy data acquisition. Consequently, we assess whether the learned policy maintains stable support distributions and suppresses harmful extrapolation beyond the data manifold (Lange et al., 2012). For each benchmark environment, a behavior policy is trained with DDPG under a fixed set of hyperparameters. The trained policy is rolled out without exploration noise to collect 1M state-action transitions, which are stored in a replay buffer. This database serves as the sole source of supervision for imitation learning. BCQ and FQL are trained on this replay buffer under identical hyperparameters, thus performance differences reflect the effect of the constraint alone. Since FQL requires the critic network to evaluate action perturbations along orthonormal basis vectors $v$, this process may induce transient instability in the initial stages. To examine this effect, we introduce an augmented variant, FQL-V, in which the cVAE is trained on a background dataset supplemented with the complete set of orthonormal bases $V$.

In Figure 4, FQL achieves superior performance relative to most of the behavior policies. The results indicate that the frictional constraint is effective across most environments. This observation is consistent with the geometric interpretation that perturbations in the normal direction dominate the extrapolation error outside the tangent space. In contrast, the performance of BCQ largely remains at the level of the behavior policies. Our frictional constraint regulates off-support perturbations while preserving behavior-policy imitation. Taken together, the proposed method does not merely fit the empirical action distribution; it also re-

shapes the sensitivity of the value function to orthonormal bases in the action space. Lastly, FQL-V yields additional gains in the early stages of selected tasks. This indicates an environment-dependent trade-off between constraint coverage and stability, wherein initial instability in the critic is mitigated. These results demonstrate that FQL improves upon BCQ and that the frictional constraint functions reliably in practice, even when the replay buffer is restricted to a fixed dataset. These performance gains arise from improved geometric regularization of constraints rather than from increased data coverage.

## 5.3. Offline RL

We compare FQL with IQL (Kostrikov et al., 2021) on the D4RL benchmark (Fu et al., 2020), a strong and widely adopted baseline for offline RL. Although our method is conceptually derived from BCQ, our offline variant does not employ a perturbation network, which is designed to produce actions that remain close to the support of the dataset. Instead, we incorporate a behavior cloning (BC) objective, following a design choice from TD7 (Fujimoto et al., 2023). As reported in Appendix D.2, offline adaptation yields encouraging results. In particular, FQL+BC demonstrates performance comparable to IQL on the HalfCheetah and Hopper tasks. However, FQL performs substantially worse on Walker2d, where training becomes notably unstable.

## 5.4. Ablation

Table 2 reports ablation results on MuJoCo benchmarks. All results denote the final average return after 1M environment timesteps, averaged over five random seeds. The default configuration includes Total Correlation (TC), a latent dimension of $\dim(\mathcal{M}) = 2\dim(\mathcal{A})$, $\arg\min Q$-based candidate orthonormal basis selection (Algorithm 1), and a replay buffer size of $N = 1M$. Each ablation modifies a single component while retaining all other settings.

*Table 2.* Average return of final evaluation over ablation variants and environments. The best-performed value is marked in **bold**.

| Ablation | Hopper-v4 | HalfCheetah-v4 | Walker2d-v4 | Ant-v4 | Humanoid-v4 |
|---|---|---|---|---|---|
| Default | $\mathbf{3370.5 \pm 220.4}$ | $11429.0 \pm 451.0$ | $4600.2 \pm 425.8$ | $\mathbf{5102.1 \pm 578.2}$ | $\mathbf{5317.3 \pm 53.7}$ |
| *Total Correlation* w/o TC | $\mathbf{3370.5 \pm 220.4}$ | $10513.8 \pm 882.0$ | $4235.9 \pm 744.2$ | $3204.6 \pm 1018.3$ | $5110.5 \pm 374.4$ |
| *Latent Dimension* $\dim(\mathcal{M}) = 1$ | $2759.3 \pm 944.5$ | $10096.0 \pm 658.9$ | $4201.0 \pm 613.5$ | $3367.2 \pm 760.8$ | $4957.4 \pm 618.4$ |
| *Action Selection* $v_i \sim \text{Uniform}(V)$ | $3219.3 \pm 702.8$ | $9267.0 \pm 1358.4$ | $\mathbf{4729.1 \pm 494.6}$ | $3160.1 \pm 797.9$ | $5006.4 \pm 482.1$ |
| *Replay Buffer Size* $N = 0.1\text{M}$ | $2271.2 \pm 740.9$ | $\mathbf{11939.1 \pm 897.4}$ | $2965.0 \pm 1548.4$ | $3611.1 \pm 776.8$ | $4715.7 \pm 753.3$ |

Removing TC generally degrades performance, suggesting that TC facilitates the separation of structured latent representations. Reducing the latent dimension to $\dim(\mathcal{M}) = 1$ consistently lowers returns across all environments, indicating that an overly restrictive latent space fails to capture essential action-relevant structures. Conversely, $\dim(\mathcal{M}) = 2\dim(\mathcal{A})$ strikes an effective balance between representation capacity and regularization (Appendix A.1). Furthermore, replacing the proposed $\arg\min Q$-based candidate selection with uniform sampling yields weaker overall results. This confirms that critic-guided selection is a vital component of FQL, enabling the agent to select informative background samples rather than arbitrary normal directions. Similarly, reducing the replay buffer size degrades performance in most environments, demonstrating that FQL benefits from a diverse replay distribution; a larger buffer improves state-action coverage and stabilizes value estimation (Fedus et al., 2020). We examine the sensitivity of FQL to the weighting coefficient $\beta$ (Higgins et al., 2017). Performance exhibits moderate sensitivity to this parameter, with no uniformly optimal value across all tasks. This suggests that while the regularization controlled by $\beta$ is beneficial, its ideal magnitude is environment-dependent (Appendix D.3).

## 6. Conclusion

We propose Frictional Q-Learning to address extrapolation error by inducing an action manifold through the explicit comparison of supported and unsupported action variations. By employing the contrastive variational autoencoder, the method provides a practical and scalable approach for continuous control. Analogous to static friction, our algorithm introduces a stability mechanism that suppresses off-support perturbations while allowing policy improvement along supported directions. This perspective is particularly relevant in practice, where complete and isotropic coverage of the action space is rarely attainable.

Our analysis also provides empirical support for the intended geometric interpretation of the method. As shown in Appendix D.4, the constructed background samples exhibit a high degree of approximate orthogonality to action samples in the learned manifold space, even without an explicit orthogonality constraint. This suggests that the proposed construction captures directions that are distinct from the primary action-relevant manifold directions. We believe that this behavior is induced by the structure of our cVAE: target actions are reconstructed with a salient component, whereas background actions are modeled under the assumption that the salient component is absent. This encourages the salient component of background actions to stay near zero, thereby disentangling target and background components and empirically yielding latent directions that are close to orthogonal.

At the same time, the current framework does not explicitly enforce isometry between the action space and the latent manifold, nor does it guarantee orthogonality by design. The observed near-orthogonality should therefore be understood as an emergent property induced by the model structure, rather than as a formal geometric guarantee. These remain important limitations of the present approach, and could be addressed in future work by integrating methods for isometric or geometry-preserving representation learning (Gropp et al., 2020; Lee et al., 2022). In addition, a more systematic study of the regularization coefficient $\beta$ in the contrastive variational autoencoder would further clarify the applicability of the method across diverse tasks. More broadly, we hope this work provides a useful foundation for future reinforcement learning methods that explicitly exploit geometric structure.

## Acknowledgements

This research work was funded by the National Research Foundation (NRF) grant funded by the Korea government (MSIT) (No. RS-2026-25477293). This work was supported by Hyundai Motor Chung Mong-Koo Foundation.

## Impact Statement

This paper presents a Reinforcement Learning methodology. The work is evaluated exclusively in simulated control environments and does not involve human subjects, personal data, or deployment. As such, the potential societal and ethical impacts of this work are consistent with those commonly associated with advances in Machine Learning, and we do not foresee any immediate adverse consequences that warrant special consideration here.

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

# A. Proofs

## A.1. Action Geometry

At state $s$, the encoder maps actions to latent vectors $u := E(s, a)$ and $c := E(s, v)$, where the generated action $v$ satisfies $a^\top v = 0$. If the map $a \mapsto E(s, a)$ acts as an isometry for a fixed state $s$, the inner product preservation implies $\langle u, c \rangle = \langle a, v \rangle = 0$. However, learned encoders often distort angles due to non-linearities and latent dimensionality. For geometric analysis, we assume $c$ remains orthogonal to $u$ and analyze the distribution relative to the data manifold. Encoded replay buffer actions concentrate near a smooth $k$-dimensional manifold $M_\mathcal{B}$ around $u$, with $1 \le k < d$. At $u$, the tangent space is $T_u M_\mathcal{B}$ and the normal space is $N_u M_\mathcal{B} := (T_u M_\mathcal{B})^\perp$. Let $P_T$ and $P_N$ denote orthogonal projections onto these spaces. The vector $c$ decomposes into $c_T := P_T c$ and $c_N := P_N c$.

**Manifold analysis.** For the trivial case $k = 1$, $T_u M_\mathcal{B} = \text{span}\{u\}$ and $N_u M_\mathcal{B} = u^\perp$. Here, the condition $c^\top u = 0$ necessitates $c \in N_u M_\mathcal{B}$. When the action manifold satisfies $k \ge 2$, the constraint $c^\top u = 0$ does not imply that $c$ is orthogonal to the entire tangent space. However, since the pair $(s, a)$ constitutes a supported sample from the replay buffer, its representation $u$ serves as the dominant reference direction in the local tangent space. While $c$ is strictly orthogonal to this dominant direction $u$, it may still possess components along other auxiliary tangent directions $(T_u M_\mathcal{B} \cap u^\perp)$. We analyze the expected energy distribution of these components through a symmetry model.

**Lemma A.1.** *Assume $u \in T_u M_\mathcal{B}$ and the latent direction $c$ is a random vector such that $\|c\| = 1$ and $c^\top u = 0$. If the distribution of $c$ is uniform on the unit sphere in $u^\perp$, then*

$$\mathbb{E}\|c_T\|^2 = \frac{k-1}{d-1}, \qquad \mathbb{E}\|c_N\|^2 = \frac{d-k}{d-1}, \qquad \frac{\mathbb{E}\|c_T\|^2}{\mathbb{E}\|c_N\|^2} = \frac{k-1}{d-k}.$$

*Proof.* Since $u \in T_u M_\mathcal{B}$, we have $P_T u = u$. The orthogonality $c^\top u = 0$ implies

$$c_T^\top u = (P_T c)^\top u = c^\top (P_T u) = c^\top u = 0.$$

Thus, $c_T$ must lie in the intersection $T_u M_\mathcal{B} \cap u^\perp$. Since $u$ occupies one dimension of the $k$-dimensional tangent space, this intersection has dimension $k - 1$. Similarly, $N_u M_\mathcal{B}$ is orthogonal to $T_u M_\mathcal{B}$, so $N_u M_\mathcal{B} \subset u^\perp$. This yields the orthogonal decomposition of the $(d-1)$-dimensional subspace $u^\perp$ where the dimensions are $k - 1$ and $d - k$, respectively:

$$u^\perp = (T_u M_\mathcal{B} \cap u^\perp) \oplus N_u M_\mathcal{B}.$$

By the uniformity assumption, the law of $c$ is rotationally invariant within $u^\perp$. Fix an orthonormal basis $\{e_1, \ldots, e_m\}$ of $u^\perp$ where $m = d - 1$. We write $c = \sum_{i=1}^m \xi_i e_i$. Since $\|c\|^2 = 1$, the sum $\sum_{i=1}^m \xi_i^2$ equals 1. Rotational invariance implies identical distribution for all coordinates. Expectation of the sum yields

$$\sum_{i=1}^m \mathbb{E}[\xi_i^2] = 1 \quad \Longrightarrow \quad m\, \mathbb{E}[\xi_1^2] = 1 \quad \Longrightarrow \quad \mathbb{E}[\xi_i^2] = \frac{1}{m} \quad \text{for all } i.$$

The squared norm of the projection onto any subspace of $u^\perp$ equals the sum of $\xi_i^2$ over the basis vectors for that subspace. For the tangent component $c_T \in T_u M_\mathcal{B} \cap u^\perp$ with dimension $k - 1$, and similarly for normal component $c_N \in N_u M_\mathcal{B}$ with dimension $d - k$, the expected energies are

$$\mathbb{E}\|c_T\|^2 = (k-1) \cdot \frac{1}{d-1} = \frac{k-1}{d-1}, \quad \mathbb{E}\|c_N\|^2 = (d-k) \cdot \frac{1}{d-1} = \frac{d-k}{d-1}.$$

*Remark* A.2. Lemma A.1 indicates that for the normal component to dominate the tangential component ($\mathbb{E}\|c_N\|^2 > \mathbb{E}\|c_T\|^2$), the latent dimension must satisfy $d > 2k - 1$. Furthermore, when $d$ is sufficiently large relative to $k$, the simple orthogonality condition $c^\top u = 0$ increasingly ensures that $c$ becomes orthogonal to the entire tangent space. In our implementation, however, the encoder is state-conditioned. Because the state dimension typically far exceeds the action dimension $(\dim(s) \gg \dim(\mathcal{A}))$, the encoder input becomes exceedingly high-dimensional. To ensure effective projection and representation learning despite this practical constraint, we experimentally set the latent dimension to approximately twice the original action dimension $(k \approx 2\dim(\mathcal{A}))$. This setting represents a necessary experimental limitation that balances the geometric preference for a large $k$ with the capacity required to encode high-dimensional state information.

### A.2. Bound

Let $\gamma \in [0, 1)$ with a bounded reward function $|r(s, a, s')| \leq R_{\max}$ and define the total variation distance as $\mathrm{TV}(p, q) :=$ $\frac{1}{2} \|p - q\|_1$. Then, the extrapolation error is defined as the difference between the action-value functions induced by the true transition dynamics $p_0(\cdot|s, a)$ and those induced by the replay buffer, denoted by $p_\theta(\cdot|s, a)$, and it admits a uniform upper bound by the discrepancy between the respective transition dynamics:

$$\|\mathcal{E}_\theta\|_\infty \leq \frac{2R_{\max}}{(1 - \gamma)^2} \sup_{s,a} \mathrm{TV}\big(p_0(\cdot|s, a), p_\theta(\cdot|s, a)\big).$$

*Proof.* From the Bellman equations $Q_0^\pi = T_0^\pi Q_0^\pi$ and $Q_\theta^\pi = T_\theta^\pi Q_\theta^\pi$, their difference can be written recursively as

$$\mathcal{E}_\theta = \gamma P_0^\pi \mathcal{E}_\theta + \sum_{s'} (p_0 - p_\theta)(s'|s, a)\big(r(s, a, s') + \gamma V_\theta^\pi(s')\big) =: \gamma P_0^\pi \mathcal{E}_\theta + b(s, a).$$

Since the value function satisfies $\|V_\theta^\pi\|_\infty \leq R_{\max}/(1-\gamma)$, it follows that $\|r + \gamma V_\theta^\pi\|_\infty \leq R_{\max}/(1-\gamma)$. For any bounded function $f$ and distributions $p, q$, the total variation distance satisfies $|\mathbb{E}_p[f] - \mathbb{E}_q[f]| \leq 2 \|f\|_\infty \mathrm{TV}(p, q)$. By the standard total variation inequality, we obtain a uniform upper bound on the bias term $b$:

$$\|b\|_\infty \leq \frac{2R_{\max}}{1 - \gamma} \sup_{s,a} \mathrm{TV}\big(p_0(\cdot|s, a), p_\theta(\cdot|s, a)\big).$$

The extrapolation error can be reformulated as $\mathcal{E}_\theta = (I - \gamma P_0^\pi)^{-1} b$. With $\|P_0^\pi\|_\infty = 1$, the Neumann series expansion yields a uniform upper bound on $(I - \gamma P_0^\pi)^{-1}$. Therefore, the extrapolation error satisfies the uniform upper bound

$$\|\mathcal{E}_\theta\|_\infty \leq \|(I - \gamma P_0^\pi)^{-1}\|_\infty \|b\|_\infty \leq \Big\| \sum_{t=0}^\infty (\gamma P_0^\pi)^t \Big\|_\infty \|b\|_\infty \leq \frac{2R_{\max}}{(1 - \gamma)^2} \sup_{s,a} \mathrm{TV}\big(p_0(\cdot|s, a), p_\theta(\cdot|s, a)\big).$$

The bound separates the extrapolation error into two components: (i) a local mismatch between transition dynamics, quantified by the total variation distance, and (ii) temporal amplification over the horizon, which induces the factor $(1-\gamma)^{-2}$. As a consequence, even a small off-support perturbation may accumulate substantial error in long-horizon (Xi et al., 2021).

### A.3. Convergence

Let $A(s)$ denote the finite set of candidate actions generated at state $s$. The set consists of actions supported by the replay buffer as well as normal direction perturbations introduced by design. We define the restricted Bellman operator

$$(TQ)(s, a) = r(s, a) + \gamma \mathbb{E}_{s' \sim p_{\mathcal{B}}(\cdot|s,a)} \Big[ \max_{a' \in A(s')} Q(s', a') \Big].$$

Since $A(s)$ is finite for all $s$ and the reward function is bounded, the operator $T$ is well-defined and bounded. Moreover, the maximization over a finite action set is non-expansive under the supremum norm, which implies that, for any two action-value functions $Q_1$ and $Q_2$, $T$ is a $\gamma$-contraction mapping and admits a unique fixed point $Q^\star$:

$$\|TQ_1 - TQ_2\|_\infty \leq \gamma \|Q_1 - Q_2\|_\infty.$$

The update of the critic can be written in the stochastic approximation form, where $(s_t, a_t, r_t, s_{t+1})$ is sampled from the replay buffer and the learning rates $\{\alpha_t\}$ satisfy the Robbins-Monro conditions:

$$Q_{t+1}(s_t, a_t) = (1 - \alpha_t)Q_t(s_t, a_t) + \alpha_t \Big[ r_t + \gamma \max_{a' \in A(s_{t+1})} Q_t(s_{t+1}, a') \Big].$$

Under the assumptions that (i) the state-action space is finite, (ii) each state-action pair is updated infinitely often, (iii) the learning rates satisfy $\sum_t \alpha_t = \infty$ and $\sum_t \alpha_t^2 < \infty$, and (iv) rewards are bounded, the update satisfies the conditions of the stochastic approximation (Singh et al., 2000). Consequently, $\Delta_t = Q_t - Q^\star$ converges to zero with probability one, and $Q_t$ converges almost surely to the unique fixed point $Q^\star$.

# B. Hyperparameters

Tables 3 and 4 summarize the common and environment-specific hyperparameters used in our experiments. The latent dimension of the cVAE, dim($\mathcal{M}$), is set to twice the action dimension dim($\mathcal{A}$). For SAC, TD3, and DDPG, we adopt the hyperparameter settings from Cleanrl (Huang et al., 2022) (https://github.com/vwxyzjn/cleanrl), whereas for MEOW, we use the authors' official implementation (https://github.com/ChienFeng-hub/meow).

*Table 3.* Common hyperparameters of FQL used in all experiments

| Parameter | Value |
| --- | --- |
| Shared optimizer | Adam (Kingma, 2014) |
| Policy Learning rate | $3 \times 10^{-4}$ |
| Discount factor | 0.99 |
| Replay buffer size | $10^6$ |
| Number of hidden layers (All networks) | 2 |
| Number of hidden units per layer (Actor & Critic) | 256 |
| Network Bias (cVAE) | False |
| Number of samples per minibatch | 256 |
| Nonlinearity | ReLU |
| Target update interval | 2 |
| Gradient steps | 1 |

*Table 4.* Environment-specific hyperparameters of FQL

| Environment | State Dim | Action Dim | Critic Lr | cVAE Lr | cVAE Dim | Beta |
| --- | --- | --- | --- | --- | --- | --- |
| Hopper-v4 | 11 | 3 | $1 \times 10^{-3}$ | $3 \times 10^{-4}$ | 256 | 2.0 |
| HalfCheetah-v4 | 17 | 6 | $3 \times 10^{-4}$ | $1 \times 10^{-3}$ | 256 | 1.0 |
| Walker2d-v4 | 17 | 6 | $1 \times 10^{-3}$ | $3 \times 10^{-4}$ | 512 | 2.0 |
| Ant-v4 | 105 | 8 | $3 \times 10^{-4}$ | $1 \times 10^{-3}$ | 256 | 2.0 |
| Humanoid-v4 | 348 | 17 | $3 \times 10^{-4}$ | $1 \times 10^{-3}$ | 512 | 1.0 |

# C. Overview

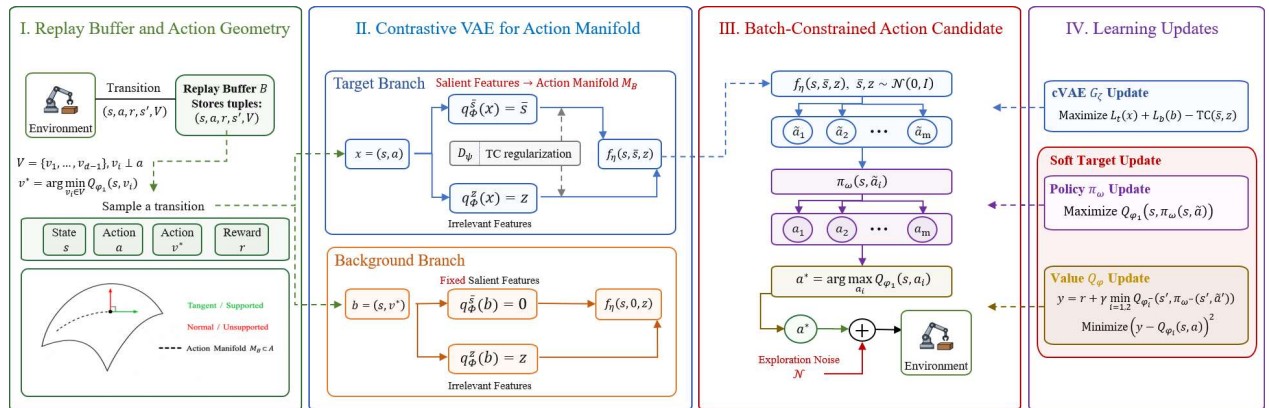

*Figure 5.* Frictional Q-Learning characterizes local action geometry from replay-buffer transitions and uses background actions to distinguish support-aware action structure from off-support variation. A contrastive VAE generates support-aware action proposals, which are refined by the policy and selected by the critic. The rightmost panel summarizes the corresponding updates.

# D. Experiments

## D.1. Manipulation

We report the performance of FQL and baseline methods on the DMControl (Tassa et al., 2018) and MyoSuite (Caggiano et al., 2022) benchmarks in Table 5. For DMControl, we report the average score across the last evaluation episodes for each task, whereas for MyoSuite, we report the average success rate over 10 tasks. All methods were trained for 1M environment timesteps over 5 seeds. To ensure a fair comparison, all methods were evaluated under a unified set of hyperparameters across tasks as much as possible. Overall, FQL achieves competitive or superior performance across most DMControl tasks and remains effective on the broader manipulation benchmarks, indicating that its benefits extend beyond locomotion.

*Table 5.* Comparison of algorithms on DMControl and MyoSuite Benchmarks

| Benchmark | Task | FQL (Ours) | TD3 | DDPG | SAC | MEOW |
|---|---|---|---|---|---|---|
| DMControl | finger-spin | $\mathbf{767.3 \pm 168.6}$ | $575.5 \pm 359.2$ | $632.3 \pm 195.6$ | $611.5 \pm 31.4$ | $434.4 \pm 14.5$ |
| | finger-turn-easy | $685.8 \pm 179.0$ | $522.0 \pm 154.3$ | $406.7 \pm 181.2$ | $\mathbf{824.4 \pm 48.2}$ | $465.7 \pm 81.7$ |
| | finger-turn-hard | $588.6 \pm 230.8$ | $354.8 \pm 85.1$ | $381.1 \pm 61.8$ | $\mathbf{717.5 \pm 106.2}$ | $418.2 \pm 116.6$ |
| | reacher-easy | $\mathbf{978.1 \pm 2.8}$ | $961.9 \pm 14.7$ | $843.7 \pm 119.2$ | $974.4 \pm 6.0$ | $962.8 \pm 15.3$ |
| | reacher-hard | $962.4 \pm 9.9$ | $949.7 \pm 45.0$ | $758.0 \pm 126.8$ | $963.7 \pm 4.7$ | $\mathbf{970.5 \pm 2.4}$ |
| | ball-in-cup-catch | $\mathbf{975.5 \pm 2.7}$ | $967.5 \pm 3.2$ | $949.8 \pm 12.1$ | $968.4 \pm 2.3$ | $957.8 \pm 10.6$ |
| Myosuite | 10-task-average | $\mathbf{30.80}\%$ | $\mathbf{31.00}\%$ | $1.8\%$ | $10\%$ | $2.8\%$ |

## D.2. Offline RL

Table 6 compares FQL+BC with IQL on the D4RL locomotion benchmarks. Results are reported on the Medium, Medium-Replay, and Medium-Expert datasets for HalfCheetah, Hopper, and Walker2d. All results denote the final normalized return after 1M training timesteps, averaged over 10 random seeds. The strong performance on HalfCheetah and competitive results on Hopper indicate that the method retains potential in the offline regime, while the weaker performance on Walker2d suggests that additional design refinement is needed for improved robustness and consistency.

*Table 6.* Comparison of FQL+BC and IQL on D4RL benchmarks. Results are reported as normalized returns.

| Environment | Dataset | FQL+BC | IQL |
|---|---|---|---|
| HalfCheetah | Medium | 50.66 | 47.41 |
| | Medium-Replay | 46.23 | 43.44 |
| | Medium-Expert | 85.25 | 88.80 |
| | **Total** | **182.14** | **179.65** |
| Hopper | Medium | 62.52 | 67.05 |
| | Medium-Replay | 73.33 | 88.46 |
| | Medium-Expert | 90.29 | 73.16 |
| | **Total** | **226.14** | **228.67** |
| Walker2d | Medium | 44.60 | 80.88 |
| | Medium-Replay | 54.87 | 66.36 |
| | Medium-Expert | 56.81 | 109.76 |
| | **Total** | **156.28** | **257.00** |
| **Total** | | **564.56** | **665.32** |

### D.3. Beta

Table 7 reports the performance of FQL under different values of the coefficient $\beta$. All results denote the final average return after 1M environment timesteps, averaged over 5 random seeds. The results indicate that performance is moderately sensitive to this parameter, and no single value is uniformly optimal across all environments. In particular, $\beta = 2$ achieves the best performance on Hopper-v4 and Ant-v4, while $\beta = 1$ performs best on HalfCheetah-v4. These results suggest that the regularization effect controlled by $\beta$ is beneficial, although its optimal strength may vary across tasks.

*Table 7.* Performance comparison across different $\beta$ values. Bold values indicate the highest average performance for each benchmark.

| Beta | Hopper-v4 | HalfCheetah-v4 | Ant-v4 |
|---|---|---|---|
| $\beta = 0$ | $2973.0 \pm 665.9$ | $13497.0 \pm 2841.6$ | $5014.0 \pm 1456.2$ |
| $\beta = 1$ | $2592.7 \pm 1382.0$ | $\mathbf{15969.6 \pm 554.2}$ | $3657.8 \pm 3069.5$ |
| $\beta = 2$ | $\mathbf{3370.5 \pm 220.4}$ | $15671.5 \pm 571.2$ | $\mathbf{6176.6 \pm 207.7}$ |
| $\beta = 5$ | $2682.2 \pm 830.2$ | $15771.1 \pm 452.5$ | $5509.7 \pm 975.5$ |

### D.4. Orthogonality

Table 8 presents the angular separation between salient components of background samples $v_i$ and action samples $a_i$ in manifold space, computed from the replay buffer of Humanoid-v4. A large fraction of samples lies near $90°$, indicating approximate orthogonality between the two sets. Although the ratio decreases as the margin becomes narrower, the overall pattern remains consistent. This result supports the intended geometric interpretation of the proposed background sample construction.

*Table 8.* Distribution of the angular separation in manifold space between background samples $v_i$ and the action samples $a_i$.

| Margin of Angle | Cosine Similarity | Average Ratio |
|---|---|---|
| $90° \pm 10°$ $(80° \sim 100°)$ | $\lvert\cos\theta\rvert < 0.1736$ | $68.14\% \pm 0.05\%$ |
| $90° \pm 5°$ $(85° \sim 95°)$ | $\lvert\cos\theta\rvert < 0.0872$ | $38.12\% \pm 0.04\%$ |
| $90° \pm 3°$ $(87° \sim 93°)$ | $\lvert\cos\theta\rvert < 0.0523$ | $23.44\% \pm 0.05\%$ |
| $90° \pm 1°$ $(89° \sim 91°)$ | $\lvert\cos\theta\rvert < 0.0175$ | $7.94\% \pm 0.05\%$ |

