# OpenReview forum: "Frictional Q-Learning"
_ICML.cc/2026/Conference — ICML 2026 regular_

### Official Review · Reviewer_fQon · 2026-03-05

**Soundness:** 3
**Presentation:** 3
**Significance:** 3
**Originality:** 4
**Overall Recommendation:** 4
**Confidence:** 3

**Summary:**

This paper proposes Frictional Q-Learning (FQL), an off-policy deep RL algorithm for continuous control that models the replay buffer as a low-dimensional action manifold and uses a contrastive variational autoencoder (cVAE) to decompose actions into tangent (supported) and normal (unsupported) directions, constraining the policy to stay near buffer-supported actions while penalizing orthogonal extrapolation via a physics-inspired friction analogy.  FQL achieves SOTA on MuJoCo benchmarks (Walker2D, Humanoid), outperforming BCQ/TD3+BC in off-policy/offline settings through stable value estimation and robust exploration.


3) Soundness*
3: good

4) Presentation*
3: good

5) Significance*
3: good

6) Originality*
4: excellent

9) Overall Recommendation*
4: Weak accept

10) Confidence*
3

11) Ethical Review Flag
No

12) Ethics Expertise Needed
N/A

**Code of Conduct Acknowledgement**
I abide by ICML code of conduct. [ppl-ai-file-upload.s3.amazonaws](https://ppl-ai-file-upload.s3.amazonaws.com/web/direct-files/attachments/16198717/a6f2be99-0996-4843-a409-b92045a104c0/22397_Frictional_Q_Learning.pdf)

**Compliance With Llm Reviewing Policy:**

Affirmed.

**Key Questions For Authors:**

1. No theoretical analysis: regret bounds or convergence to manifold? Would elevate soundness.
2. cVAE \(\beta\) task-sensitive; auto-tuning method? Boosts significance.
3. Orthogonal selection via \(\min Q\): ablate random/uniform? Confirms mechanism.

**Limitations:**

Yes (task-specific \(\beta\), no theory); standard impact.

**Strengths And Weaknesses:**

**Soundness**: Good. Geometric manifold intuition sound; cVAE ELBO + TC term standard. Expts comprehensive (MuJoCo, offline ablations, 5 seeds). Ablations validate components (critic-based vs. aug-based orthogonal). No theory (e.g., regret/convergence), empirical only.

**Presentation**: Good. Clear physics analogy (Fig. 1); Alg. 1 repro.; baselines standard. Dense notation; cVAE eqs. could clarify latent roles.

**Significance**: Good. Addresses extrapolation in off-policy RL; SOTA on high-dim tasks timely for robotics/offline RL. Modular cVAE constraint practical.

**Originality**: Excellent. Novel friction manifold + dual tangent/normal constraints via cVAE; orthonormal orthogonal basis fresh geometric view.

---

> ### Author Rebuttal · Authors · 2026-03-31
>
> We sincerely thank the reviewer for the insightful and detailed feedback. We address each point below and will revise the manuscript accordingly. \
> All experiments in rebuttal were conducted for 1M environment steps with 5 random seeds.
>
> > **Q1: Theoretical Analysis**
>
> We agree that incorporating regret bounds or convergence guarantees would strengthen our theoretical foundation. Our current framework primarily provides a geometric justification for identifying normal directions rather than a full convergence analysis. We will explicitly clarify this scope and state this limitation in the revised manuscript. Extending this analysis to establish formal guarantees, particularly regarding extrapolation error and manifold-constrained policy improvement, remains a vital direction for future work.
>
> > **Q2: $\beta$ sensitivity**
>
> We acknowledge the cVAE’s sensitivity to $\beta$ and our reliance on grid search. This choice is necessitated by $\beta$’s complex interaction with both latent regularization and the contrastive structure, which makes standard auto-tuning less reliable than in vanilla batch-constrained VAEs. As shown in R.Table 1, while moderate $\beta$ values generally yield better performance, no single value is universally optimal. We will clarify this in the revision and explore adaptive $\beta$ schedules in future work.
>
> > **R.Table 1**
> |Beta|Hopper-v4|HalfCheetah-v4|Ant-v4|
> |---|---|---|---|
> |$\beta=0$|2973.0±665.9|13497.0±2841.6|5014.0±1456.2|
> |$\beta=1$|2592.7±1382.0|**15969.6±554.2**|3657.8±3069.5|
> |$\beta=2$|**3370.5±220.4**|15671.5±571.2|**6176.6±207.7**|
> |$\beta=5$|2682.2±830.2|15771.1±452.5|5509.7±975.5|
>
> > **Q3: Orthogonal selection mechanism**
>
> We thank the reviewer for the suggestion and agree that comparing the critic-guided selection against random sampling is essential to validate our mechanism. While the orthogonal complement determines the number of background directions, their specific instantiation is a key design choice. As shown in R.Table 2, an ablation comparing critic-guided selection ($v_i = \arg\min_v Q(s, v)$) with uniform sampling demonstrates that the former improves performance across most benchmarks. We will include this analysis in the revision to clarify the critic’s role in our selection process.
>
> > **R.Table 2**
> |Algorithm|Hopper-v4|HalfCheetah-v4|Walker2d-v4|Ant-v4|Humanoid-v4|
> |---|---|---|---|---|---|
> |FQL ($v_i = \arg\min_v Q(s, v)$)|**3370.5±220.4**|**11429.0±451.0**|4600.2±425.8|**5102.1±578.2**|**5317.3±53.7**|
> |FQL ($v_i\sim\text{Uniform}(V)$)|3219.3±702.8|9267.0±1358.4|**4729.1±494.6**|3160.1±797.9|5006.4±482.1|

---

> > ### Author Rebuttal · Reviewer_fQon · 2026-04-03
> >
> > The writer addressed all the claims

---

> > > ### Author Response · Authors · 2026-04-06
> > >
> > > We sincerely appreciate the reviewer’s suggestion, which has significantly strengthened the rigor of the paper. We hope that our responses and additional results have adequately addressed the concerns raised, and if so, we would be grateful if this could be reflected in the assessment.

---

### Official Review · Reviewer_qPUz · 2026-03-08

**Soundness:** 2
**Presentation:** 3
**Significance:** 2
**Originality:** 3
**Overall Recommendation:** 4
**Confidence:** 4

**Summary:**

This paper introduces Frictional Q-Learning (FQL), an off-policy continuous-control method designed to mitigate extrapolation error by restricting policy improvement to actions that remain close to replay-buffer support. The key idea is to model replay-supported actions as lying on a low-dimensional action manifold, with supported tangent directions separated from unsupported normal directions using a contrastive VAE. The method is evaluated primarily on online MuJoCo locomotion benchmarks, with additional fixed-buffer offline/imitation experiments to test whether the learned constraint improves robustness to off-support action perturbations.

**Compliance With Llm Reviewing Policy:**

Affirmed.

**Final Justification:**

The authors addressed several of my concerns and provided additional experimental analysis to strengthen the paper.

**Key Questions For Authors:**

- How sensitive is FQL to the environment-specific settings in Table 3, and can the method work with a more unified hyperparameter configuration?
- Can the authors add non-locomotion tasks, such as manipulation domains, to test whether the gains are specific to locomotion?
- Can the authors provide mechanism-focused diagnostics, such as value smoothness or perturbation analyses relative to the learned action manifold?
- Does Table 2 apply only to FQL, or also to reproduced baselines? How were baseline learning rates selected?
- How does performance change as replay-buffer size or support coverage is varied?
- Why were BEAR and BAIL omitted from the benchmark comparisons despite being discussed in related work?

**Limitations:**

Yes

**Strengths And Weaknesses:**

- The paper is clearly written, and the focus on mitigating extrapolation error from weak replay support is important and well motivated. The static-friction analogy is an appealing way to organize the geometric intuition. The empirical study is reasonably broad within continuous-control locomotion, including online comparison against DDPG, TD3, SAC, and MEOW, plus additional offline/imitation-style comparisons against BCQ on fixed replay buffers. However, I think the evaluation could still be strengthened through better hyperparameter transparency, broader domain coverage, and more direct analyses of the claimed geometric mechanism.
- An area for improvement is hyperparameter consistency. Table 2 lists common hyperparameters, including a shared actor learning rate of $3×10^{−4}$, while Table 3 introduces environment-specific FQL settings for critic LR, cVAE LR, cVAE dimension, and $\beta$. At the same time, the paper says baselines were reproduced using Stable-Baselines3 and official implementations. It is therefore unclear which hyperparameters were shared across methods and which followed baseline defaults. The paper would be stronger if it explicitly stated whether baseline-specific default settings were used and how the actor learning rate in Table 2 was chosen.
- Relatedly, FQL appears to rely on nontrivial environment-specific tuning. A stronger empirical case would come from either showing robustness to these settings or demonstrating competitive performance with a more nearly uniform hyperparameter configuration across domains. This matters especially because the margins in Table 1 are mixed: FQL is strongest on Walker2D and Humanoid, while on Hopper, Ant, and HalfCheetah it is closer to or below the best baseline.
- Task diversity could also be expanded. The benchmarks are all Gymnasium MuJoCo locomotion environments, which leaves open whether the method’s benefits depend on the repetitive structure of gait-like dynamics. I would like to see non-locomotion domains as well, for example manipulation benchmarks such as MetaWorld, or at least a discussion of whether the proposed geometric constraints are expected to transfer beyond locomotion.
- Additionally, the paper still does not directly validate the proposed mechanism as much as it could. The imitation-learning section is a useful step in this direction, but more direct qualitative and quantitative analysis would help: for example, analyzing critic smoothness, action distributions, or sensitivity to off-manifold perturbations along test trajectories, ideally including a low-dimensional toy example where the tangent/normal interpretation can be visualized explicitly.
- I would also ask the authors to discuss why BEAR and BAIL are included in related work but omitted from the experimental comparisons, especially since they are conceptually closer to the paper’s offline/support-constrained framing than some of the online baselines.
- Finally, it would be useful to study sensitivity to replay-buffer size or dataset coverage, since replay support is central to the method and the current setup fixes the replay buffer size at $10^6$. That would help test whether FQL’s advantage grows specifically in lower-coverage settings, which is where its motivation is strongest.
- A related line of work that would be worth discussing is replay-buffer augmentation and interpolation, including mixup-style data augmentation methods [1-4].

**References:**

[1] Wang, Kaixin, et al. "Improving generalization in reinforcement learning with mixture regularization." Advances in Neural Information Processing Systems 33 (2020): 7968-7978.

[2] Lin, Junfan, et al. "Continuous transition: Improving sample efficiency for continuous control problems via mixup." 2021 IEEE International Conference on Robotics and Automation (ICRA). IEEE, 2021.

[3] Sander, Ryan, et al. "Neighborhood mixup experience replay: Local convex interpolation for improved sample efficiency in continuous control tasks." Learning for Dynamics and Control Conference. PMLR, 2022.

[4] Sinha, Samarth, Ajay Mandlekar, and Animesh Garg. "S4rl: Surprisingly simple self-supervision for offline reinforcement learning in robotics." Conference on Robot Learning. PMLR, 2022.

---

> ### Author Rebuttal · Authors · 2026-03-31
>
> We sincerely thank the reviewer for the insightful and detailed feedback. We address each point below and will revise the manuscript accordingly. \
> Except for the newly reported R.Table 1, all experiments in rebuttal were conducted for 1M environment steps with 5 random seeds.
>
> > **Q1: Hyperparameter transparency**
>
> We acknowledge the lack of clarity regarding hyperparameter sharing. The revision will explicitly separate FQL-specific, common, and baseline settings (e.g., SB3/official), clarifying that Table 2 refers only to FQL. While baselines follow official defaults (e.g., $10^{-3}$ for DDPG/SAC/MEOW, $3 \times 10^{-4}$ for TD3), we adopted a $3 \times 10^{-4}$ learning rate for FQL to stabilize incremental updates. This protocol will be detailed in the revised manuscript and appendix.
>
> > **Q2: Unified Hyperparameters**
>
> We thank the reviewer for the feedback. To align with standard practices, we will update our main comparison to report final evaluation returns rather than average maximum returns. Under this metric, FQL remains highly competitive or superior. Additionally, R.Table 1 demonstrates FQL’s robustness under a unified hyperparameter configuration across all tasks. While environment-specific tuning can further optimize performance, these results confirm that FQL does not strictly depend on aggressive per-task tuning to outperform several baselines.
>
> > **R.Table 1**
> |Algorithm|Hopper-v4|HalfCheetah-v4|Walker2D-v4|Ant-v4|Humanoid-v4|
> |---|---|---|---|---|---|
> |FQL(Tuned)|**3370.5±220.4**|**15969.6±554.2**|**5659.9±510.1**|6176.6±207.7|**6437.2±722.1**|
> |FQL(Unified)|3196.7±576.9|15726.5±407.2|5069.9±1253.9|5054.8±1249.0|6097.7±1029.0|
> |SAC|2548.4±850.1|15476.8±244.6|4815.5±1698.4|3098.5±3690.3|5682.6±328.0|
> |MEOW|3190.3±222.2|7241.6±3722.4|1919.2±2524.4|5635.1±356.8|4650.2±1168.1|
> |TD3|**3371.9±181.8**|15120.7±517.6|5104.4±787.8|**6542.3±310.1**|5373.5±1879.9|
> |DDPG|1824.1±1061.2|15628.1±981.5|1744.7±436.6|-237.8±507.4|2618.1±749.6|
>
> > **Q3: Task Diversitiy**
>
> To address concerns regarding task diversity, we expanded our evaluation to non-locomotion benchmarks, including DMControl (manipulation) and MyoSuite (musculoskeletal control), with unified hyperparameters. While we agree with the MetaWorld suggestion, the limited rebuttal period led us to prioritize these representatives. As shown in R.Table 2, FQL demonstrates strong performance on DMControl and an encouraging 30.8% average success rate across 10 MyoSuite tasks. These findings empirically confirm that FQL’s geometric constraints generalize effectively beyond gait-like locomotion dynamics.
>
> > **R.Table 2**
> |Benchmark|Task|FQL|
> |---|---|---|
> |DMControl(Score)|Finger Spin|767.3±168.6|
> ||Finger Turn(Easy)|685.8±179.0|
> ||Finger Turn(Hard)|588.6±230.8|
> ||Reacher(Easy)|978.1±2.8|
> ||Reacher(Hard)|962.4±9.9|
> ||Ball-in-cup Catch|975.5±2.7|
> |MyoSuite(Success Rate)|10-Task Average|30.8%|
>
> > **Q4: Mechanism-focused diagnostics**
>
> We agree that, beyond return-based evaluations, direct diagnostics are necessary to validate the proposed geometric mechanism. The revision will analyze critic-guided normal perturbations against tangential and random ones regarding value and action support. Notably, R.Table 3 confirms that target and background representations concentrate near orthogonality on Humanoid-v4 (we measured the angle between salient representations of $a_i$ and background samples $v_i$), achieving the intended geometric separation. Combining with training value evolution plots demonstrating critic stability, these diagnostics will provide direct empirical support for our proposed mechanism.
>
> > **R.Table 3**
> |Margin of Angle(Degree)|Threshold(Cosine)|Average Ratio(±Std)|
> |---|---|---|
> |90°±5°(85°~95°)|abs<0.0872|38.1%±0.0%|
> |90°±3°(87°~93°)|abs<0.0523|23.4%±0.1%|
> |90°±1°(89°~91°)|abs<0.0175|7.9%±0.1%|
>
> > **Q4: Replay Buffer**
>
> Given the centrality of replay support, we examined buffer sensitivity by reducing capacity from 1M to 0.1M (R.Table 4). While smaller buffers generally degrade performance, the non-monotonicity observed across environments underscores that coverage quality is as vital as raw size. We will include this analysis in the revision and expand our related work to distinguish FQL’s geometric constraints from buffer interpolation methods (e.g., mixup).
>
> > **R.Table 4**
> |Algorithm|Hopper-v4|HalfCheetah-v4|Walker2d-v4|Ant-v4|Humanoid-v4|
> |---|---|---|---|---|---|
> |FQL(N=1M)|**3370.5±220.4**|11429.0±451.0|**4600.2±425.8**|**5102.1±578.2**|**5317.3±53.7**|
> |FQL(N=0.1M)|2271.2±740.9|**11939.1±897.4**|2965.0±1548.4|3611.1±776.8|4715.7±753.3|
>
> > **Q5: BEAR and BAIL**
>
> We acknowledge the relevance of BEAR and BAIL. We prioritized BCQ as a structurally identical (VAE-based) baseline to isolate the impact of our geometric constraints. In contrast, BAIL lacks a VAE structure, and BEAR differs primarily in its loss formulation rather than architecture. The revision will include a broader theoretical and empirical comparison with both methods.

---

> > ### Author Rebuttal · Reviewer_qPUz · 2026-04-03
> >
> > I thank the authors for their detailed responses, the additional analysis will strengthen the quality of the paper. For the newly added tasks, the final version should include comparisons to the discussed baseline algorithms. If all these changes are applied to the final draft, I'm happy to raise my score.

---

> > > ### Author Response · Authors · 2026-04-06
> > >
> > > We truly agree that the suggested additional analyses strengthen the paper, and we will integrate R.Tables 1--4 and the clear hyperparameter table into the final draft. Also, we will include a figure comparing the smoothness of the value function for our method and the baselines. In addition, R.R Table 1 reports the total average reward of our method and the baselines on the newly added manipulation benchmarks. Due to the limited rebuttal window, we prioritized deterministic baselines for the newly added suites (e.g., MyoSuite). We regret that we were unable to include stochastic baselines at this stage, as they typically require substantially greater computational cost and could not be completed within the available timeframe. We will include the full results, including stochastic baselines, in the camera-ready version. We believe these additional results also help address the concern regarding unified hyperparameters, as they show that the same overall setup remains applicable in new environments. More broadly, they provide both empirical basis and limitations for further assessing the robustness of FQL across a wider range of benchmarks.
> > >
> > > > **R.R Table 1**
> > > | Benchmark | Task | FQL (Ours) | TD3 | DDPG | SAC | MEOW |
> > > | :--- | :--- | :--- | :--- | :--- | :--- | :--- |
> > > | **DMControl(Score)** | Finger Spin | **$767.3 \pm 168.6$** | $575.5 \pm 359.2$ | $632.3 \pm 195.6$ | $611.5 \pm 31.4$ | $434.4 \pm 14.5$ |
> > > | | Finger Turn(Easy) | $685.8 \pm 179.0$ | $522.0 \pm 154.3$ | $406.7 \pm 181.2$ | **$824.4 \pm 48.2$** | $465.7 \pm 81.7$ |
> > > | | Finger Turn(Hard) | $588.6 \pm 230.8$ | $354.8 \pm 85.1$ | $381.1 \pm 61.8$ | **$717.5 \pm 106.2$** | $418.2 \pm 116.6$ |
> > > | | Reacher(Easy) | **$978.1 \pm 2.8$** | $961.9 \pm 14.7$ | $843.7 \pm 119.2$ | $974.4 \pm 6.0$ | $962.8 \pm 15.3$ |
> > > | | Reacher(Hard) | $962.4 \pm 9.9$ | $949.7 \pm 45.0$ | $758.0 \pm 126.8$ | $963.7 \pm 4.7$ | **$970.5 \pm 2.4$** |
> > > | | Ball-in-cup Catch | **$975.5 \pm 2.7$** | $967.5 \pm 3.2$ | $949.8 \pm 12.1$ | $968.4 \pm 2.3$ | $957.8 \pm 10.6$ |
> > > | **MyoSuite(Success Rate)** | 10-Task Average | 30.80% | 31.00\% | 1.8% | 10% | - |
> > > >
> > >
> > > We sincerely appreciate the reviewer’s suggestion, which has significantly strengthened the rigor of the paper. We hope that our responses and additional results have adequately addressed the concerns raised, and if so, we would be grateful if this could be reflected in the assessment.

---

### Official Review · Reviewer_AUG8 · 2026-03-13

**Soundness:** 4
**Presentation:** 3
**Significance:** 3
**Originality:** 3
**Overall Recommendation:** 5
**Confidence:** 2

**Summary:**

This paper addresses the challenge of extrapolation error in off-policy reinforcement learning through a novel geometric perspective. The authors model the replay buffer support as a smooth, low-dimensional action manifold and propose that extrapolation errors primarily arise from deviations along the manifold's normal directions. Drawing an analogy to static friction in classical mechanics, the paper introduces a stability condition that bounds value sensitivity without excessive dependence on the discount factor $\gamma$. To implement this, the authors develop Frictional Q-Learning (FQL), which utilizes a Contrastive Variational Autoencoder (cVAE) to disentangle supported (tangential) variations from unsupported (normal) ones. The method is evaluated on MuJoCo benchmarks and demonstrates competitive performance, particularly in high-dimensional tasks.

**Compliance With Llm Reviewing Policy:**

Affirmed.

**Final Justification:**

The explanations were helpful for understanding the paper. I am keeping the score

**Key Questions For Authors:**

- Did you monitor the Jacobian of the encoder during training? If the local isometry property doesn't hold, does the cVAE still successfully separate normal from tangential directions?

- How does the affine transformation (used to handle bounded action spaces)  affect the theoretical guarantees of the tangent/normal space decomposition near the boundaries of the action space?

- Could you provide a performance comparison against IQL or Diffusion-QL on the D4RL benchmark?

- What happens if the background samples $v_i^*$ in the cVAE are chosen randomly instead of selecting the "least-supported" actions via the critic? This would help clarify the importance of the critic's role in the contrastive objective.

**Limitations:**

yes

**Strengths And Weaknesses:**

Strengths:

- The geometric interpretation of extrapolation error as a "friction" phenomenon on an action manifold is original and provides a fresh intuition for batch-constrained RL.

- Unlike many prior works where error bounds scale quadratically with $(1-\gamma)^{-2}$, the proposed stability condition (Equation 5) offers a more robust theoretical foundation for long-horizon tasks.

- The transition from the friction analogy to a concrete cVAE architecture is well-motivated. The use of an orthonormal basis for the orthogonal complement to identify normal directions is a clever technical contribution.

- The method shows strong performance on challenging environments like Walker2d-v4 and Humanoid-v4, and the imitation learning experiments effectively validate the manifold-support claim.

- The appendix provides comprehensive hyperparameters and implementation details.

Weaknesses:

- The theoretical justification for identifying normal directions (Theorem 4.1) heavily relies on the assumption that the encoder is a local isometry ($J_a^\top J_a \approx I$). In practice, deep neural networks rarely satisfy this property without explicit regularization (e.g., spectral normalization or specific Jacobian penalties). The paper lacks an empirical verification or a discussion on how FQL behaves when this assumption is violated.

- While the friction analogy is a strong motivational tool, it remains slightly unclear if the "anisotropy ratio" provides benefits beyond what a standard manifold-constrained approach (like a well-tuned VAE or Diffusion model) would achieve. More ablation studies on the "friction threshold" logic would be beneficial.

- The empirical evaluation primarily compares FQL against older baselines such as BCQ, BEAR, and CQL. To fully demonstrate state-of-the-art (SOTA) performance, comparisons with more recent methods like IQL (Implicit Q-Learning) or Diffusion-based Offline RL (e.g., Diffuser, QQL) should be included.

- The contrastive structure relies on separating "salient" and "irrelevant" factors using Total Correlation (TC). This introduces additional complexity and potential sensitivity to the hyperparameter $\beta$ and the discriminator $D_\psi$. The paper would be stronger with a sensitivity analysis of these components.

---

> ### Author Rebuttal · Authors · 2026-03-31
>
> We sincerely thank the reviewer for the insightful and detailed feedback. We address each point below and will revise the manuscript accordingly. \
> All rebuttal experiments were conducted for 1M environment steps with 5 random seeds, but 10 seeds for the results in R.Table 3.
>
> > **Q1: Local isometry assumption**
>
> We agree that Theorem 4.1 relies on a local isometry assumption that is not explicitly enforced in our implementation. We will clarify in the revision that this assumption is used only as an analytical device for characterizing normal directions, rather than as a strict claim about the behavior of deep encoders in practice.
>
> > **Q2: Anisotropy Ratio**
>
> We thank the reviewer for this suggestion. While our goal was to show that the anisotropy ratio imposes a stronger directional constraint than standard manifold-constrained models through imitation learning, we agree that the specific contribution of the friction-threshold logic requires further isolation. We will clarify this in the revision and include additional comparisons between our threshold mechanism and standard manifold-constrained approaches.
>
> > **Q3: Comparisons with recent baselines**
>
> We agree that comparing with recent offline RL methods, such as IQL and diffusion models, would strengthen our evaluation. While we initially focused on baselines most directly related to our formulation, we recognize the importance of these contemporary methods for a complete assessment. We will broaden the discussion and include additional comparisons in the revised manuscript.
>
> > **Q4: Total Correlation (TC)**
>
> We agree that the TC term increases optimization complexity and sensitivity to regularization and the auxiliary discriminator. We therefore conducted a sensitivity analysis (R.Table 1) and will discuss this trade-off explicitly in the revision.
>
> > **R.Table 1**
> |Algorithm|Hopper-v4|HalfCheetah-v4|Walker2d-v4|Ant-v4|Humanoid-v4|
> |---|---|---|---|---|---|
> |FQL(w/TC)|**3370.5±220.4**|**11429.0±451.0**|**4600.2±425.8**|**5102.1±578.2**|**5317.3±53.7**|
> |FQL(w/oTC)|3370.5±220.4|10513.8±882.0|4235.9±744.2|3204.6±1018.3|5110.5±374.4|
>
> > **Q5: Isometry assumption**
>
> Although local isometry is not explicitly enforced, we tested whether the cVAE still separates normal and tangential information in practice. Specifically, on Humanoid-v4, we measured the angle between salient representations of $a_i$ and background samples $v_i$. As shown in R.Table 2, these representations are strongly biased toward orthogonality, suggesting that the intended geometric separation emerges empirically even without explicit Jacobian control.
>
> > **R.Table 2**
> |Margin of Angle(Degree)|Threshold(Cosine)|Average Ratio(±Std)|
> |---|---|---|
> |90°±5°(85°~95°)|abs<0.0872|38.1%±0.0%|
> |90°±3°(87°~93°)|abs<0.0523|23.4%±0.1%|
> |90°±1°(89°~91°)|abs<0.0175|7.9%±0.1%|
>
> > **Q6: Affine Transformation**
>
> The affine transformation is a practical tool for asymmetric action spaces, not a core theoretical component. It maps actions to a symmetric space for orthonormal basis construction and is then inverted before environment interaction. Since our main benchmarks already have symmetric action spaces, this transformation was not used in the main experiments, ensuring our empirical results are free from associated boundary effects.
>
> > **Q7: D4RL Benchmark**
>
> We thank the reviewer for this suggestion. To assess whether FQL extends beyond our main setting, we evaluated a behavior-cloning variant on D4RL (R.Table 3). FQL+BC is competitive with or outperforms IQL on HalfCheetah and Hopper, but underperforms on Walker2d. These results are encouraging but not conclusive, suggesting that the core idea transfers to offline RL while further refinement is needed for consistent performance across datasets.
>
> > **R.Table 3**
> |Environment|Dataset|FQL+BC|IQL|
> |---|---|---|---|
> |HalfCheetah|Medium|**50.7**|47.4|
> ||Medium-Replay|**46.2**|43.4|
> ||Medium-Expert|85.3|**88.8**|
> ||*Total*|**182.1**|179.7|
> |Hopper|Medium|62.5|**67.1**|
> ||Medium-Replay|73.3|**88.5**|
> ||Medium-Expert|**90.3**|73.2|
> ||*Total*|226.1|**228.7**|
> |Walker2d|Medium|44.6|**80.9**|
> ||Medium-Replay|54.9|**66.4**|
> ||Medium-Expert|56.8|**109.8**|
> ||*Total*|156.3|**257.0**|
> |**D4RL Total**||564.6|**665.3**|
>
> > **Q8: Orthogonal selection mechanism**
>
> We agree that comparing critic-guided selection with random sampling is important for clarifying the critic’s role. As shown in R.Table 4, critic-guided selection improves performance on most benchmarks, except Walker2d in these runs. We will include this ablation to show that the critic helps identify informative background samples.
>
> > **R.Table 4**
> |Algorithm|Hopper-v4|HalfCheetah-v4|Walker2d-v4|Ant-v4|Humanoid-v4|
> |---|---|---|---|---|---|
> |FQL($v_i=\arg\min_v Q$)|**3370.5±220.4**|**11429.0±451.0**|4600.2±425.8|**5102.1±578.2**|**5317.3±53.7**|
> |FQL($v_i\sim\text{Uniform}$)|3219.3±702.8|9267.0±1358.4|**4729.1±494.6**|3160.1±797.9|5006.4±482.1|

---

> > ### Author Rebuttal · Reviewer_AUG8 · 2026-04-05
> >
> > Thank you for the rebuttal. The explanations were helpful for understanding the paper. I will maintain my current score.

---

### Official Review · Reviewer_ezKn · 2026-03-13

**Soundness:** 2
**Presentation:** 2
**Significance:** 2
**Originality:** 3
**Overall Recommendation:** 4
**Confidence:** 3

**Summary:**

The authors propose Frictional Q-learning, an off-policy method for continuous control that aims to avoid state-action pairs not supported by the replay buffer. To achieve this, actions are mapped to a low-dimensional manifold. Directions that stay on the manifold (tangential directions) correspond to actions that are supported by the data, while directions outside the manifold, especially the normal directions that are perpendicular, are discouraged. The mapping is learned using a contrastive variational autoencoder (VAE). It treats actions within the data support as positive samples and unsupported actions as negative samples. The method is inspired by concepts from mechanics. In particular, it draws an analogy to static friction, which counteracts forces. Similarly, extrapolation error counteracts updates that move the policy toward convergence. In this sense, the friction acts as a penalty that keeps the policy close to the data distribution. Experiments on the MuJoCo benchmark show competitive performance.

**Compliance With Llm Reviewing Policy:**

Affirmed.

**Final Justification:**

After considering both the paper and the authors’ rebuttal, I maintain my recommendation. The paper’s main strength is its originality: the friction-inspired geometric view is novel, intuitive, and likely to be of interest to future work on off-policy RL. The method also appears technically reasonable, and the rebuttal helped clarify several of my main questions, including the policy definition, manifold dimensionality, and the role of key hyperparameters. At the same time, I still have some reservations. Clarity remains a weakness, as parts of the paper are difficult to follow, and it is hard to judge from the rebuttal alone how much the revised presentation will improve. I also share concerns raised in the discussion about hyperparameter transparency and the overall strength and breadth of the empirical validation. The rebuttal improved these points and added useful analysis, but some aspects still remain mixed for me.

**Key Questions For Authors:**

1. How does the (low) dimensionality of the manifold affect the stability of the method?
2. The replay buffer action distribution is approximated by the generative network. How does approximation error influence convergence?
3. How does the hyperparameter beta of the contrastive VAE affect learning behavior?
4. The definition of the policy is not entirely clear to me. Is it simply a procedure that maximizes the learned Q-values? Does it output an action in the original action space or in the low-dimensional action manifold? From the definition in Section 4.5 it seems to be the latter, while the pseudocode suggests that the learned policy operates in the original action space. Could the authors clarify this?

**Limitations:**

Yes

**Strengths And Weaknesses:**

Strengths:

* Interesting analogy from mechanics used to motivate the method
* The approach has solid mathematical foundations and seems novel and a good foundation for future methods

Weaknesses:

* Some parts are difficult to follow. A large overview figure showing all components and their interactions would have been very helpful
* The results are somewhat mixed e.g. TD3 appears to outperform the method on Hopper and Walker
* The method requires training an additional generative network, which may introduce notable computational overhead.

Remarks:
* In Figure 2, the unit vector u initially appears to be perpendicular to both t and n. Only after reading the text carefully it becomes clear that this is not the case. It would help to clarify this more clearly in the figure or caption.

---

> ### Author Rebuttal · Authors · 2026-03-31
>
> We sincerely thank the reviewer for the insightful and detailed feedback. We address each point below and will revise the manuscript accordingly. \
> Except for the newly reported R.Table 1, all experiments in rebuttal were conducted for 1M environment steps with 5 random seeds.
>
> > **Q1: Large Overview Figure**
>
> We agree that an overview figure enhances readability and will include a pipeline diagram in the revised manuscript, illustrating the interactions between the cVAE, Critic, and Actor networks.
>
> > **Q2: Mixed Results**
>
> While our initial report of average maximum returns showed mixed results, we have updated our metrics to final performance to align with standard research practices. As shown in R.Table 1, this change demonstrates a clearer performance gap in favor of FQL. We will include these final metrics in the revision and, space permitting, clarify the distinction between the two evaluation methods.
>
> > **R.Table 1**
> |Algorithm|Hopper-v4|HalfCheetah-v4|Walker2D-v4|Ant-v4|Humanoid-v4|
> |---|---|---|---|---|---|
> |FQL|**3370.5±220.4**|**15969.6±554.2**|**5659.9±510.1**|6176.6±207.7|**6437.2±722.1**|
> |SAC|2548.4±850.1|15476.8±244.6|4815.5±1698.4|3098.5±3690.3|5682.6±328.0|
> |MEOW|3190.3±222.2|7241.6±3722.4|1919.2±2524.4|5635.1±356.8|4650.2±1168.1|
> |TD3|**3371.9±181.8**|15120.7±517.6|5104.4±787.8|**6542.3±310.1**|5373.5±1879.9|
> |DDPG|1824.1±1061.2|15628.1±981.5|1744.7±436.6|-237.8±507.4|2618.1±749.6|
>
> > **Q3: Unit Vector $u$**
>
> To avoid misinterpreting unit vector $u$ as orthogonal to both $t$ and $n$, we will update the figure caption to clarify its geometric relationship: “Note that $u(\theta)$ lies entirely within the 2D plane $\mathcal{P}$ spanned by the tangent vector $t$ and normal vector $n$. The angle $\theta$ explicitly denotes the rotation of $u$ within coplanar space”.
>
> > **Q4: Dimensionality Of The Action Manifold**
>
> As discussed in Appendix A, the manifold dimension $dim(\mathcal{M})$ balances expressiveness and regularization. Low dimensions can lead to under-expressiveness and model collapse, while excessively high dimensions weaken the manifold constraint. We will clarify this intuition in the revised text and include R.Table 2 to demonstrate that a moderate manifold dimension strikes an optimal balance.
>
> > **R.Table 2**
> |Algorithm|Hopper-v4|HalfCheetah-v4|Walker2d-v4|Ant-v4|Humanoid-v4|
> |---|---|---|---|---|---|
> |FQL(dim(M)=2dim(A))|**3370.5±220.4**|**11429.0±451.0**|**4600.2±425.8**|**5102.1±578.2**|**5317.3±53.7**|
> |FQL(dim(M)=1)|2759.3±944.5|10096.0±658.9|4201.0±613.5|3367.2±760.8|4957.4±618.4|
>
> > **Q5: Approximation Error**
>
> Theoretically, In the ideal case where $p_{\mathcal{B}} = p_{\mathcal{M}}$ suppresses extrapolation error, but generative approximation errors can weaken this constraint and cause actions to deviate from the replay distribution. Such deviations increase the likelihood of selecting out-of-distribution actions, hindering stable policy improvement. To mitigate this, we introduced more stringent batch-constraints with frictional analogy. Our imitation learning experiments confirm that a tighter approximation of the replay distribution leads to improved downstream performance compared to standard BCQ.
>
> > **Q6: Hyperparameter $\beta$**
>
> $\beta$ balances reconstruction and latent regularization in the contrastive VAE. While prior ELBO-based derivations omit an explicit $\beta$ weight, we found empirical tuning essential for performance. As shown in R.Table 3, $\beta$ presents a task-dependent trade-off, with moderate values like $\beta=2.0$ yielding the most consistent results across environments. We will incorporate this discussion in the revised manuscript.
>
> > **R.Table 3**
> |Beta|Hopper-v4|HalfCheetah-v4|Ant-v4|
> |---|---|---|---|
> |Beta=0.0|2973.0±665.9|13497.0±2841.6|5014.0±1456.2|
> |Beta=1.0|2592.7±1382.0|**15969.6±554.2**|3657.8±3069.5|
> |Beta=2.0|**3370.5±220.4**|15671.5±571.2|**6176.6±207.7**|
> |Beta=5.0|2682.2±830.2|15771.1±452.5|5509.7±975.5|
>
> > **Q7: The Definition Of The Policy**
>
> We thank the reviewer for identifying the inconsistency in Section 4.5 and Algorithm 1. The policy indeed outputs actions in the original action space, and we have updated the manuscript to accurately define its operation space. This correction significantly improves the paper’s mathematical clarity. The corrected formulations are as follows:
>
> Algorithm 1: $\omega \leftarrow \arg\max_\omega \sum\bigl(Q_{\varphi_1}(s, \pi_{\omega}(s, \tilde{a}))\bigr)$ \
> Section 4.5: $\pi(s) = \arg\max_{\tilde{a}} Q_{\varphi_1}(s, \pi_{\omega}(s,\tilde{a})), \quad \tilde{a} \sim \{f_{\eta}(s)\}_{i=1}^m$

---

> > ### Author Rebuttal · Reviewer_ezKn · 2026-04-04
> >
> > I thank the authors for their responses. This answered some of my concerns. However, based on the written comments alone, I still cannot fully judge how much the clarity of the paper has improved. I also agree with the reviewer’s concern regarding hyperparameter transparency: it is important to ensure that the comparisons are truly fair and that the proposed method was not tuned more carefully than the baselines. That said, I still find the analogy interesting, and the results now appear somewhat clearer, even though the training curves still leave me with mixed impressions. For these reasons, I will maintain my current assessment.

---

### Decision · Program_Chairs · 2026-04-30

**Decision:**

Accept (regular)

**Comment:**

The paper introduces an original off-policy method that addresses extrapolation error by modeling replay buffer support as a low-dimensional action manifold. The core merit lies in using a contrastive VAE to decompose actions into tangent (supported) and normal (unsupported) directions, providing a physics-inspired "friction" analogy to penalize updates that deviate from the data distribution. Rebuttal results strengthened the submission by demonstrating competitive performance on high-dimensional Humanoid and Walker2d tasks, as well as generalization to manipulation and musculoskeletal domains using a unified hyperparameter configuration. However, the authors must implement several essential changes suggested by the reviewers to ensure technical clarity and reproducibility. Specifically, the final version must include a large pipeline diagram illustrating the interactions between the cVAE, critic, and actor networks to improve readability. Additionally, the authors must explicitly correct the policy operation space definitions in Section 4.5 and Algorithm 1 to resolve the mathematical inconsistencies identified by the reviewers regarding whether the policy outputs actions in the original or manifold space.